# COMPARISON VISUAL INSTRUCTION TUNING

## ABSTRACT

Comparing two images in terms of Commonalities and Differences (CaD ) is a fundamental human capability that forms the basis of advanced visual reasoning and interpretation. It is essential for the generation of detailed and contextually relevant descriptions, performing comparative analysis, novelty detection, and making informed decisions based on visual data. However, surprisingly, little attention has been given to these fundamental concepts in the best current mimic of human visual intelligence - Large Multimodal Models (LMMs). We develop and contribute a new two-phase approach CaD-VI for collecting synthetic visual instructions, together with an instruction-following dataset CaD-Inst containing 349K image pairs with CaD instructions collected using CaD-VI . Our approach significantly improves the CaD spotting capabilities in LMMs, advancing the SOTA on a diverse set of related tasks by up to 17.5%. It is also complementary to existing difference-only instruction datasets, allowing automatic targeted refinement of those resources increasing their effectiveness for CaD tuning by up to 10%. Additionally, we propose an evaluation benchmark with 7.5K open-ended QAs to assess the CaD understanding abilities of LMMs.

## 1 INTRODUCTION

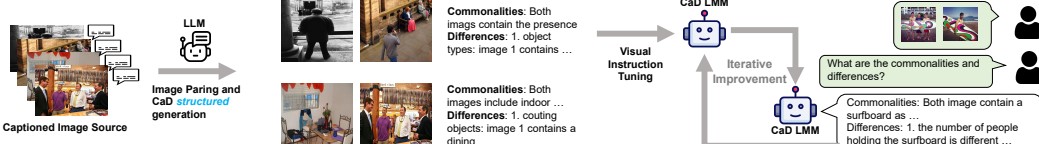

Figure 1: CaD-VI concept. We collect and pair densely captioned source images to form synthetic CaD instructions using an LLM. The resulting synthetic CaD Visual Instruction dataset is used to train the first CaD enabled LMM that is in turn used in iterative self-refinement by annotating new paired images from additional sources using the CaD LMM, and re-training the model with a growing and more comprehensive CaD-Inst dataset (contributed in this work).

Understanding the Commonalities and Differences (CaD ) between two signals (e.g., images) is a basic capability innate to humans (IxDF, 2016). Spotting change and difference alerts us to interesting events happening in our surroundings, warns us of hazard, and drives us toward learning new concepts exposed after the change or relative movement. Understanding what is common helps structure visual information and allows differences to emerge by elimination. Together, these form powerful tools for human learning and acquiring world knowledge.

The forefront of modern AI shifted with the recent emergence of foundation Large Language Models (LLMs) (Bommasani et al., 2022), where the top-performing ones (et al., 2024b;a; Anthropic, 2024; AI@Meta, 2024) closely align to human reasoning and world-knowledge capabilities. LLMs' great performance and wide applicability quickly led to their wide adoption into most of the current ML pipelines. In the Vision community, this impacted the development of Large Multi-modal Models (LMMs) (Liu et al., 2023b; Yang et al., 2023; et al., 2024a; Huang et al., 2023; Li et al., 2023b; Dong et al., 2024; Sun et al., 2023a) largely considered the best available mimic of human visual intelligence to date. While multiple methods for adding multi-modal support to LLMs have been proposed, currently the more popular and better performing open LMMs largely rely on tuning using Visual Instructions (VI) (Liu et al., 2023b; Zhu et al., 2023b). These methods align image tokens produced by visual encoders to be 'understandable' by an LLM decoder, allowing images to be seamlessly integrated into the LLM decoder input context stream together with the query text during

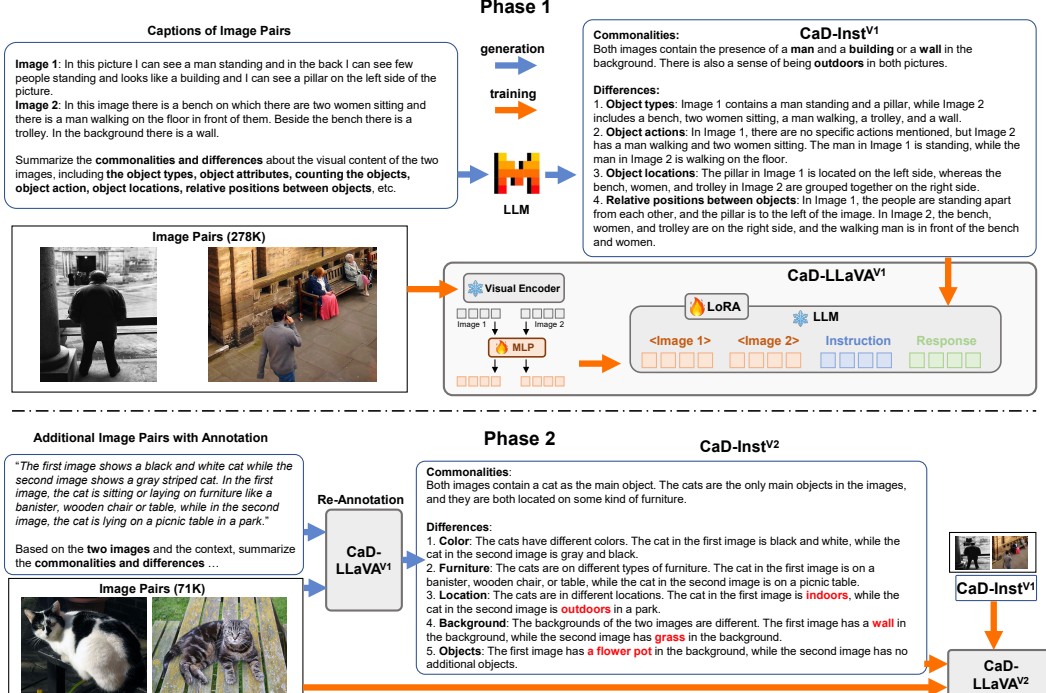

Figure 2: Pipeline of our two-phase CaD-VI : In Phase-1, we leverage captions for image pairs and an LLM to generate CaD VI data - CaD-Inst$^{V1}$ (278K), and perform visual instruction tuning on it to arrive at the Phase-1 model CaD-LLaVA$^{V1}$ . In Phase-2, we leverage CaD-LLaVA$^{V1}$ to generate CaD VI data on additional image pairs and collect CaD-Inst$^{V2}$ (71K). Visual instruction tuning with CaD-Inst$^{V1}$ and CaD-Inst$^{V2}$ leads to our final model CaD-LLaVA$^{V2}$ .

inference. In most recent methods (Liu et al., 2023b; Huang et al., 2023; Li et al., 2023b; Dong et al., 2024), VI takes the form of a multi-turn conversation: with 'human' turns providing image context and asking the questions, and LMM turns answering them (Liu et al., 2023b). However, the majority of VI data focused on providing merely a single image in the VI conversations (Liu et al., 2023b), while only a few works included multi-image VI samples (Sun et al., 2023a; Awadalla et al., 2023), and surprisingly, very few included some form of CaD VI data (Huang et al., 2023; Li et al., 2023b;a) to enable CaD support in the resulting LMM.

Due to the fundamental importance of endowing LMMs with CaD capabilities, thus getting them closer to achieving human visual intelligence in all its diversity, we propose CaD-VI - a multi-phase CaD generation approach, for progressive dense and structured CaD VI data collection (concept shown in Fig. 1), which we employ to build CaD-Inst training curriculum and associated CaD-QA benchmark comprised of CaD-related open-ended questions, both contributed in this work. In essence, the final CaD-Inst curriculum associates diverse and large-scale (349K) image pair collection with highly detailed and structured CaD summaries. CaD summaries computed for an additional set of 7.6K image pairs, are used for extracting open CaD-related QA resulting in CaD-QA .

As shown in Fig. 2, the Phase-1 of CaD-VI is a 'cold start' where, in the absence of LMMs with substantial CaD capabilities, we leverage image captions and an LLM to hallucinate (coarse) CaD VI data - CaD-Inst$^{V1}$ (278K), where we collect *structured* and *detailed* CaD summaries for our paired images sourced from a dense & large-scale image collection (Pont-Tuset et al., 2020). Training on the first phase CaD-Inst$^{V1}$ data we arrive at CaD-LLaVA$^{V1}$ - an LMM that has strong CaD capabilities compared to a large variety of leading LMMs including the very few trained with some CaD data (see Sec. 5). Next, leveraging our CaD-LLaVA$^{V1}$ model to produce non-hallucinated, image-informed CaD data, we generate additional CaD instructions into the collection CaD-Inst$^{V2}$ (71K). Combining CaD-Inst$^{V1}$ and CaD-Inst$^{V2}$ we form CaD-Inst and train our final CaD-LLaVA$^{V2}$ 7B and 13B LMMs to achieve (1) significant (up to 17.5%) absolute improvement over a large variety of recent SOTA LMMs over a variety of 5 CaD-related existing closed-QA evaluation benchmarks (namely BISON(Hu et al., 2019), SVO Probes(Hendricks & Nematzadeh, 2021), NLVR2(Suhr et al., 2019), EQBEN(Wang et al., 2023), and COLA(Ray et al., 2023)), and (2) strong (up to over 20%)

relative improvements on our contributed open-QA CaD benchmark - CaD-QA . Additionally, as CaD-Inst can be safely mixed with the LLaVA VI data (Liu et al., 2023a), we show in Tab. 4 that our CaD-LLaVA$^{V2}$ models effectively avoid forgetting the general capabilities of the corresponding LLaVA LMMs.

Our contributions are as follows: (i) we contribute CaD-Inst - a large-scale visual instruction tuning dataset for enhancing CaD reasoning capabilities of LMMs; (ii) we contribute CaD-QA - an open QA evaluation benchmark for assessing CaD capabilities; (iii) we contribute and open source a CaD-VI methodology for collecting CaD instruction tuning data and re-purposing datasets with existing difference annotations; (iv) we demonstrate significant (up to 17.5%) improvements in CaD reasoning for LMMs trained using CaD-Inst as well as potential to scale CaD-Inst via self-improvement by CaD-Inst -trained models.

## 2    RELATED WORK

**Large Multimodal Models.** LMMs have shown significant advancements in integrating visual and textual data, enhancing the ability of deep neural networks to understand and generate multimodal content. BLIP-2 employs a bootstrapping approach that leverages frozen image encoders and large language models through a querying transformer, achieving remarkable results on various vision-language tasks with fewer parameters compared to previous models (Li et al., 2023e). Similarly, MiniGPT-4 (Zhu et al., 2023a) and LLaMA-Adapters (Zhang et al., 2023b) utilize pretrained visual and language models, with adapters aligning image tokens to language tokens, improving the efficiency and performance of multimodal understanding and generation. In addition to these early models, the LLaVA series (Liu et al., 2023b), including LLaVA 1.5 (Liu et al., 2023a) and LLaVA 1.6 (Liu et al., 2024), have enhanced visual instruction tuning, enabling better handling of single-image inputs and more accurate multimodal outputs. The InternLM XComposer 2.0 VL (Zhang et al., 2023a), EMU2 (Sun et al., 2024), Otter (Li et al., 2023b), SparklesChat (Huang et al., 2023), and MMICL (Zhao et al., 2024) extend these capabilities by incorporating multiple images as input, thereby enriching the models' understanding and generation of text based on complex visual scenes. These models showcase the evolution from single-image to multi-image inputs, highlighting the progress in multimodal learning architectures and applications.

**Visual Instruction Tuning Datasets.** The success of LMMs builds on the collection of high-quality visual instruction tuning data, either constructed from existing VQA datasets (Gong et al., 2023; Goyal et al., 2017b; Hudson & Manning, 2019; Dai et al., 2023; Li et al., 2023f), curated image-text pairs (Zhu et al., 2023a) and LLM-generated instruction-following data with input of rich human annotations (Liu et al., 2023b;a; Zhang et al., 2023c; Zhao et al., 2023; Li et al., 2023a). However, the collection of multimodal data for learning commonalities and differences between two images is still under-explored.

**Image Commonalities and Differences.** Only a few datasets contain difference-only related annotation (Jhamtani & Berg-Kirkpatrick, 2018a; Li et al., 2023a). Spot-the-diff (Jhamtani & Berg-Kirkpatrick, 2018b) collects human-annotated short change descriptions for surveillance video frames. Our CaD-Inst$^{V1}$ data collection is partially inspired by the differences-only data collection done by (Li et al., 2023a) as a small part of their VI strategy. However, different from (Li et al., 2023a) we: (i) collect both differences *and commonalities* (compared to only differences in (Li et al., 2023a)); (ii) we leverage a significantly more *dense* caption-source of (Pont-Tuset et al., 2020) compared to (Chen et al., 2015) used in (Li et al., 2023a); (iii) we are *structuring* our differences in CaD according to 6 axes (whichever applicable on case basis) - object types, attributes, counting, actions, locations, and relative positioning, also explicitly asking the LLM to extract (from the dense captions) information along these axes, while (Li et al., 2023a) produced unstructured difference description text; (iv) unlike (Li et al., 2023a) we are not relying on the existence of manually collected object bounding boxes; (v) the scale of our data is approx. 4 times larger than of (Li et al., 2023a). Due to these differences, as evident from the direct comparison in Tab. 5, training the same model on CaD-Inst$^{V1}$ has significant performance advantages over training on CaD instructions of (Li et al., 2023a). To summarize, our work focuses on CaD understanding, largely neglected by the visual instruction tuning community. We propose a new CaD-VI approach for collecting synthetic visual instructions and enhancing the CaD analysis capabilities in LMMs. CaD-VI not only advances the state-of-the-art in related tasks by significant margins but also complements existing

datasets (Jhamtani & Berg-Kirkpatrick, 2018a; Li et al., 2023a) by enabling their automatic targeted refinement, thereby improving their effectiveness for CaD tuning.

# 3 CaD-VI - Two-Phase CaD Visual Instruction Tuning

As illustrated in Fig. 2, our CaD-VI consists of two phases: in Phase-1, we employ an LLM to generate summary of CaD for image pairs (Sec. 3.1) and perform visual instruction tuning on the collected data (Sec. 3.2); in Phase-2, we leverage the Phase-1 model to generate CaD on additional image pairs and perform training with combined instruction data from both phases (Sec. 3.3).

## 3.1 Phase-1a: LLM Instruction Data Collection - CaD-Inst$^{V1}$

In our first phase, we leverage an LLM to generate a summary of commonalities and differences for a pair of two images, as shown in Fig. 2 (top row). Specifically, we construct image pairs and prompt an LLM, supplying it with two image captions (one per image) and an instruction prompt asking it to summarize all the commonalities and differences according to the provided captions, contributing to our first phase CaD instruction data collection denoted as CaD-Inst$^{V1}$ .

**Image Source.** We select the Localized Narratives dataset (Pont-Tuset et al., 2020) which consists of 873K image-caption pairs with diverse samples sourced from COCO (Lin et al., 2014; Chen et al., 2015), Flickr30K (Young et al., 2014), ADE20K (Zhou et al., 2019) and Open Images (Kuznetsova et al., 2020). The captions are generated by transcription from spoken descriptions of the image content, which are quite dense, detailed, and descriptive with an average length of 36.5 words. To cover comprehensive visual contents and increase the diversity in terms of commonalities and differences, we collect 278K image pairs with different levels of similarity between their captions. We compute similarity by counting the number of overlapping nouns in the corresponding captions.

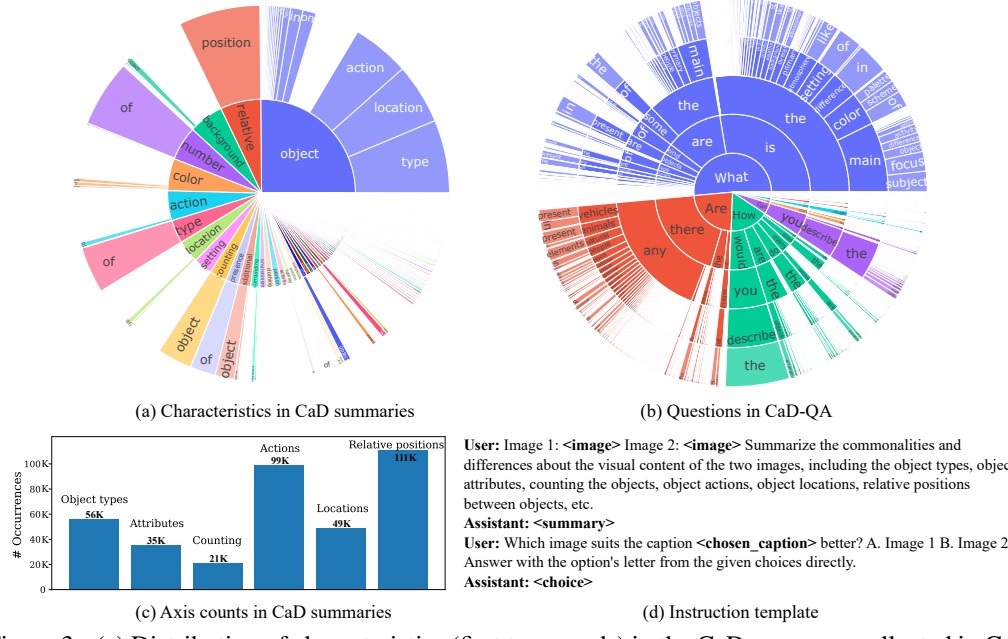

(a) Characteristics in CaD summaries

(b) Questions in CaD-QA

(c) Axis counts in CaD summaries

(d) Instruction template

Figure 3: (a) Distribution of characteristics (first two words) in the CaD summary collected in CaD-Inst$^{V1}$ ; (b) Distribution of question types (first five words) in the evaluation benchmark CaD-QA ; (c) Axis counts in CaD summaries; (d) Two-turn conversation template.

**LLM Data Generation.** In this work, we focus on employing open-source foundation models for data collection. The current open-source LMMs do not have strong capabilities of visual reasoning and instruction following when processing multiple input images. In this case, using caption as a symbolic representation of each image and employing an LLM with strong text instruction-following ability for generation of comparison summary of multiple input images is a more robust way of data collection than using open-source LMMs. The practice of this data collection pipeline with LLMs and dense captions is verified in the original LLaVA (Liu et al., 2023b) and many following works (Li et al., 2023a; Huang et al., 2023; Zhang et al., 2023c).

We leverage the Mixtral $8 \times 7$B LLM (Jiang et al., 2024) for generating detailed and structured summaries of commonalities and differences for pairs of images. As the LLM can only accept text as input, in Phase 1 we use image captions to represent visual content of images. This is a rather crude approximation, which is alleviated in Phase 2 of our CaD-VI approach. To encourage the diverse and creative generation of commonalities and differences, we do not provide in-context examples of expected output in the prompt to the LLM. Furthermore, we specifically prompt the LLM *to structure* the commonalities and differences summaries according to the following 6 visual aspects: (i) object types; (ii) attributes; (iii) counts; (iv) actions; (v) locations; and (vi) relative positions; as illustrated in Fig. 2. We provide detailed prompts in the Appendix. Importantly, LLM is not forced to produce all 6 aspects in every summary; they are generated adaptively according to the available content.

**Generated Data Statistics.** In CaD-Inst$^{V1}$ we collected structured summaries of CaD for 278K image pairs, with average length of 157 words (40 for commonalities and 117 for differences). The summaries are structured according to 6 axes, appearing unevenly on a case-to-case basis based on the LLM decision. We illustrate the distribution of data characteristics in Fig. 3(a), and the total observed axis counts in Fig. 3(c). More statistics and details are provided in the Appendix.

**CaD visual instructions data.** We construct a two-turn conversation for each image pair. In the first turn, we define the task of summarizing CaD by providing the encoded visual tokens of the two images and instructing the model to summarize the CaD , where the response part of the turn is the LLM-generated structured summary collected above. In this instruction, we do not provide the image captions, forcing the model to rely only on image tokens to complete the task. In the second turn, we reinforce the image-text alignment by employing a simple task of text-to-image retrieval to avoid forgetting the model's general capabilities. We randomly sample one of the two captions and request the model to select the image (from the current pair) to which the caption belongs. Through ablation study in Tab. 7, we show that while this task itself does not lead to satisfying results, combining it with the task of summarizing commonalities and differences results in significant improvement. The template for the two-turn conversation is illustrated in Fig. 3(d).

## 3.2 PHASE-1B: CaD VISUAL INSTRUCTION TUNING

**Architecture.** As illustrated in Fig. 2, we use our collected CaD-Inst$^{V1}$ data to perform visual instruction tuning using the open-sourced code of LLaVA-1.5 (Liu et al., 2023a) LMM. The LLaVA-1.5 model consists of $\phi_L(\cdot; \theta_L)$ - a pretrained Vicuna 1.5 (Zheng et al., 2023) LLM (finetuned from LLama 2 (Touvron et al., 2023b)); $\phi_V(\cdot; \theta_V)$ - a pretrained visual encoder CLIP ViT-L/14@336px (Radford et al., 2021); and $\phi_M(\cdot; \theta_M)$ - a two-layer MLP projector converting the visual encoder tokens to post-embedding layer LLM tokens.

Given a pair of two images $x_{V_1}, x_{V_2}$ and the instruction $x_I$, the MLP projects the visual features computed by the visual encoder into embedded language tokens, *i.e.* $v_k = \phi_M(\phi_V(x_{V_k}; \theta_V); \theta_M), k \in \{1, 2\}$. Then the projected visual features and instruction text tokens are concatenated and fed into the LLM, where the response text tokens are generated in an autoregressive manner, *i.e.*

$$\hat{x}_R^i = \phi_L([v_1, v_2, x_I, \hat{x}_R^{<i}]; \theta_L), \tag{1}$$

where $\hat{x}_R^i$ denotes the $i$-th token in the generated response.

**Training.** We finetune the LLaVA-1.5 model using the LLaVA (Liu et al., 2023b) pipeline. Specifically, following LLaVA pre-training, we finetune only the pretrained projection MLP and the (frozen) LLM with LoRA adapters (Hu et al., 2021). We minimize the CLM loss of the next token prediction in the responses:

$$\mathcal{L}_{CLM} = \sum_i - \log p(\hat{x}_R^i | V_1, V_2, x_I, x_R^{<i}) \tag{2}$$

To preserve the general VL capabilities of the LMM, we merge CaD-Inst$^{V1}$ with the finetuning data of LLaVA-1.5 (665K samples). In Tab. 4 we show that CaD-VI indeed preserves the general LMM capabilities compared to LLaVA-1.5 as evaluated on the popular SEED benchmark (Li et al., 2023d). The Phase-1 CaD visual instruction tuning results in our cold-start model CaD-LLaVA$^{V1}$ which is an LMM that can be leveraged for annotating visual commonalities and differences.

### 3.3 PHASE-2: DATA COLLECTION AND VISUAL INSTRUCTION TUNING

**Phase-2a: LMM-based CaD Instruction Collection.** While in Phase 1 we used an LLM to extract a CaD summary based on human-generated captions, for Phase 2 data collection we leverage our Phase 1 model CaD-LLaVA$^{V1}$ and additional image pairs to extract the CaD summaries informed by the images directly. Here we select the Scene-Difference (Li et al., 2023a) collection as an additional image source. It contains 71K pairs of similar images from COCO (Lin et al., 2014) and provides annotation of unstructured difference-only summaries (see Fig. 2 bottom left for an example). We feed both the image pairs and the original annotations into our CaD-LLaVA$^{V1}$ model, and generate a *structured summary* of *both* commonalities and differences. The exact prompt is provided in the Appendix. This leads to our phase-2 CaD instruction data - CaD-Inst$^{V2}$ . As shown in Tab. 5, our collected CaD instructions significantly improve over the utility of the original (Li et al., 2023a) annotations. As part of our analysis in Tab. 5 and 6, and additional experiments provided in Appendix, we also show that similarly out-of-distribution image pair collections or even unlabeled image pair collections can be effectively leveraged for our Phase-2.

In Phase-2, we generate CaD data leveraging both captions and the CaD image analysis capabilities of our Phase-1 model. This significantly reduces hallucinations and improves the quality of the Phase-2 stage CaD dataset as evident by the significant performance improvement obtained by Phase-2 model over Phase-1 model (Tab. 5 E and F). In the ablation in Sec. 6 (Tab. 6) we also show that image captions can be included in Phase-2 data collection.

In Phase-1, we have image pairs of different similarity levels while in Phase-2 we have highly similar image pairs which lead to more fine-grained difference summaries. We combine data of both phases.

**Phase-2b CaD Visual Instruction Tuning** We follow the Phase-1b introduced in Sec. 3.2 for CaD visual instruction tuning. Here we finetune on a combination of LLaVA 1.5 (Liu et al., 2023a) finetune data (665K), CaD-Inst$^{V1}$ data (278K) and CaD-Inst$^{V2}$ data (71K). This phase of CaD visual instruction tuning leads to the Phase 2 model, denoted as CaD-LLaVA$^{V2}$ .

## 4 CaD-QA - BENCHMARK OF OPEN-ENDED CaD QA

In order to evaluate the capability of LMMs on answering open-ended questions regarding commonalities and differences of a pair of two images, we construct and contribute the CaD-QA benchmark.

**Data Collection.** Similar to the data collection pipeline introduced in Sec. 3.1, we employ Visual Genome (Krishna et al., 2017) and the detailed image captions from SVIT (Zhao et al., 2023) as image & caption source. We collect 7.5K image pairs with 8 or more overlapping nouns in their captions. For each pair, we employ the Mixtral $8\times7$B LLM to produce the structured CaD summaries from the captions. Next, we prompt Mixtral with both the image captions and the CaD summary, instructing it to generate a multi-turn conversation with several rounds of Q&A, providing some in-context examples of the desired layout (see Appendix for the prompt). Finally, we randomly select one Q&A per conversation.

**Benchmark Statistics.** There are 7520 QA pairs with an average answer length of 26 words. Among these, we also include 2916 questions asking about the content of only one of the two images. It requires the precise attention of the LMM on the corresponding image to correctly answer these questions. Our CaD-QA covers diverse question types as illustrated in Fig. 3(b).

**LLM-assisted Evaluation.** Motivated by LLMs' ability to judge response quality consistently with human assessment (Zheng et al., 2023), we employ the Mixtral $8\times7$B LLM to compare the generated responses to the collected open-ended QA responses. We feed the question, correct answer, and the predicted answer into the LLM and instruct it to provide a rating between 0 and 5 for the predicted answer quality. We provide the prompt in the Appendix. In order to mitigate the bias from the the same LLM used for evaluation, we include additional evaluations with different LLMs, in-context examples of scoring cases and human study in the Appendix.

## 5 EXPERIMENTS

**Evaluation Datasets** We evaluate on several VQA benchmarks of closed-ended and open-ended questions. For **closed-ended VQA on image pairs**, we include BISON (Hu et al., 2019) and SVO Probes (Hendricks & Nematzadeh, 2021) both consisting of samples with an image pair and a text

| Dataset | # Instruction | BISON | SVO | NLVR2 | EQBEN | COLA |
| Random chance | Data | 50% | 50% | 50% | 25% | 25% |
|---|---|---|---|---|---|---|
| SparklesChat | 6.5K | 56.70% | 43.93% | 58.00% | 19.17% | 20.00% |
| Otter | 2.8M | 40.67% | 47.33% | 52.00% | 8.33% | 8.10% |
| MMICL | 5.8M | 80.00% | 88.13% | 56.67% | 20.83% | 25.71% |
| EMU2-Chat | 1.3M | 46.00% | 47.93% | 60.00% | 7.50% | 13.33% |
| InternLM-XComposer2-VL | >600K | 80.67% | 82.07% | 66.67% | 25.00% | 32.38% |
| LLaVA 1.6 7B | <1M | 66.00% | 70.40% | 58.67% | 20.83% | 11.90% |
| LLaVA 1.6 13B | <1M | 81.33% | 82.13% | 60.00% | 17.50% | 24.76% |
| LLaVA 1.5 7B | 665K | 54.00% | 46.80% | 61.33% | 17.50% | 7.62% |
| LLaVA 1.5 13B | 665K | 59.33% | 56.27% | 66.00% | 16.67% | 12.38% |
| CaD-VI 7B | 1M | 95.33% | 92.73% | 66.67% | 39.17% | 40.95% |
| CaD-VI 13B | 1M | **96.67%** | **93.00%** | **69.33%** | **42.50%** | **43.33%** |

Table 1: Performance on closed-ended VQA tasks with image pairs in accuracy. Here the method CaD-VI denotes our Phase-2 model CaD-LLaVA$^{V2}$ .

| Dataset | CaD-QA | VG comm. | VG diff. | COLA comm. | COLA diff. |
|---|---|---|---|---|---|
| SparklesChat | 3.01 | 2.41 | 3.12 | 1.52 | 1.22 |
| Otter | 2.20 | 1.88 | 1.97 | 1.37 | 0.81 |
| MMICL | 2.01 | 1.79 | 1.94 | 1.73 | 0.59 |
| EMU2-Chat | 1.20 | 1.04 | 1.08 | 1.22 | 0.41 |
| InternLM-XComposer2-VL | 2.90 | 2.08 | 2.69 | 1.72 | **1.36** |
| LLaVA 1.6 7B | 3.10 | 2.23 | 2.73 | 1.71 | 1.22 |
| LLaVA 1.6 13B | 3.19 | 2.19 | 2.69 | 1.93 | 1.01 |
| LLaVA 1.5 7B | 2.54 | 1.79 | 1.75 | 1.44 | 1.02 |
| LLaVA 1.5 13B | 2.65 | 2.16 | 2.41 | 1.57 | 1.10 |
| CaD-VI 7B | 3.29 | 2.32 | **3.85** | **2.14** | 1.25 |
| CaD-VI 13B | **3.34** | **2.58** | 3.68 | 2.13 | 1.31 |

Table 2: Performance on CaD-QA and tasks of CaD summary prediction evaluated using LLM-as-a-judge ratings (range 0 to 5). Here the method CaD-VI denotes our Phase-2 model CaD-LLaVA$^{V2}$ .

query that needs to be matched with one of the images in the pair (chance is 50%). EQBEN (Wang et al., 2023) and COLA (Ray et al., 2023) contain samples composed of a pair of two images together with the two textual descriptions. The goal is to correctly match images with corresponding texts (chance is 25%). Furthermore, we evaluate on NLVR2 (Suhr et al., 2019) which comprises samples of a pair of two images and a reasoning sentence. The task is to assess the correctness of the reasoning and has a random chance of 50%. We also evaluate SEED-Bench Video (Li et al., 2023d) with two frames sampled from the video to explore the generalization value of our CaD tuning for video understanding. SEED-Bench Video contains three partitions from SEED-Bench and has multi-choice questions on action recognition/prediction or procedure understanding with four answer options per question. For **open-ended tasks**, use the LLM-as-a-judge metric (Sec. 4). We evaluate open-ended QAs on our CaD-QA . Furthermore, we also directly evaluate the quality of LMM predicted CaD summaries for 210 image pairs in COLA with shorter summaries generated from brief captions, and for the 7.5K lengthy summaries from CaD-QA generated from detailed VG captions. More details and statistics of the datasets are provided in the Appendix.

**Implementation Details** We leverage the Mixtral $8\times7B$ Instruct v0.1 and set the maximum token size to 750 data collection and 20 for open-ended task evaluation. For visual instruction tuning, we use the official implementation of LLaVA and tune the LLaVA 1.5 7B model with LoRA. We set the batch size to 128 and LoRA learning rate for LLM and the projector is set to $1 \times 10^{-4}$ and $2 \times 10^{-5}$ correspondingly. All experiments are run on $4\times$A100 80G GPUs. More details are in Appendix.

**Comparison to State-of-the-Art LMMs**

We first compare our final model CaD-LLaVA$^{V2}$ (denoted by CaD-VI in Table) to state-of-the-art LMMs on closed-ended VQA in Table 1. SparklesChat (Huang et al., 2023), Otter (Li et al., 2023b), MMICL (Zhao et al., 2024), EMU2-Chat (Sun et al., 2023a), InternLM-Xcomposer2-VL (Zhang et al., 2023a) all include samples with multi-image inputs in the visual instruction tuning while LLaVA 1.5 (Liu et al., 2023a) and LLaVA 1.6 (Liu et al., 2024) are tuned with only single image instructions. The evaluated benchmarks are challenging due to the visually very similar image pairs with subtle compositional differences where the LMMs could easily make an incorrect decision leading to performance below random chance. Our CaD-VI 7B model already outperforms all the other baselines on the five benchmarks and our 13B finetuned model further boosts the performance. We also include the comparison to a recent video LMM in the appendix.

Image 1       Image 2

(a)

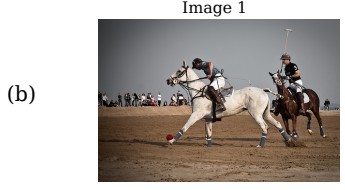 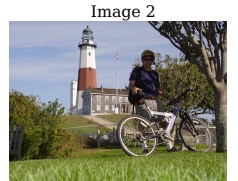

Q: What objects are present in Image 2 that are not in Image 1?

GT: In Image 2, there is a well-equipped shower, a bathtub, a commode, a mirror, a wooden shelf, and various items such as a green candle, air freshener spray, and silver faucet. These objects are not present in Image 1.

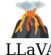
LLaVA : In Image 2, there is a toilet visible in the bathroom, which is not present in Image 1. The man is standing in front of the toilet while holding an umbrella.

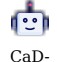
CaD-LLaVA : In Image 2, there are several objects that are not present in Image 1, including a man, a mirror, a sink, a toilet, a shower, a window, a door, and a ceiling. The man is standing in the bathroom, and the mirror, sink, toilet, and shower are all located in the same room. The window and door are also present in the bathroom, and the ceiling is visible at the top of the image.

Image 1       Image 2

(b)

Q: What is the central figure in each image?

GT: In Image 1, the central figures are several horses and their riders, engaged in a polo game. In Image 2, the central figure is a man holding a mountain bike.

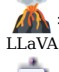
LLaVA : The central figure in each image is a man riding a bicycle.

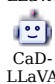
CaD-LLaVA : In Image 1, the central figure is a man riding a horse. In Image 2, the central figure is a man riding a bicycle.

Figure 4: Examples of Q&A pairs in CaD-QA together with LMM predicted answers (Red and green texts denote incorrect and correct description).

Table 2 demonstrates the comparison to the baseline LMMs on open-ended tasks of CaD-QA and of CaD summary prediction on image pairs. Our CaD-VI models outperform the baselines on four of the five open-ended tasks, with the exception of COLA difference summary where our 13B model achieves a rating (1.31) close to the best performing InternLM-XComposer2 model (1.36). We include additional evaluations with different LLMs, in-context examples of scoring cases and human study in the Appendix, which shows that the Mixtral-assisted evaluation is valid as it maintains the same ranking as when using strongest LLMs as judge.

In Fig. 4, we show examples of Q&A pairs in CaD-QA together with predicted answers from CaD-LLaVA$^{V2}$ model and the vanilla LLaVA 1.5 model. The vanilla LLaVA model has incorrect answers by either mistakenly combining the contents in two images (Fig. 4(a), *the man is standing in front of the toilet while holding an umbrella*) or attending to incorrect image (Fig. 4(b)), demonstrating lacking of capability of properly comparing two images. Our CaD-LLaVA$^{V2}$ manages to correctly differentiate between the two images, attend to the corresponding content queried and draw a summary of comparison. More qualitative results on CaD -QA and BISON can be found in the Appendix.

Furthermore, we explore whether our CaD instruction tuning improves video understanding evaluated using SEED-Bench Video in Table 3. In the evaluation setting of LLaVA, only one frame per SEED-Bench video is passed to the LMM. To explore the impact of our CaD tuning, we compare this to evaluating using two frames as input. As shown in Table 3, although multiple baseline LMMs achieve better performance in single-frame setting, our CaD-VI 13B model performs the best in the two-frame setting with a significant performance improvement of 2.93% on top of the single-frame performance. The only higher improvement is achieved by Otter, which however struggles below the 25% chance level performance. This underlines that our CaD tuning improves the temporal understanding between video frames.

| # Input Frames | 1 | 2 |
|---|---|---|
| SparklesChat | 21.81% | 19.09% (▼-2.72%) |
| Otter | 18.19% | 23.00% (▲+4.81%) |
| EMU2-Chat | 43.43% | 41.09% (▼-2.34%) |
| InternLM-XComposer2-VL | 41.07% | 40.16% (▼-0.91%) |
| LLaVA 1.6 7B | 41.95% | 42.03% (▲+0.08%) |
| LLaVA 1.6 13B | 41.85% | 41.35% (▼-0.50%) |
| LLaVA 1.5 7B | 37.43% | 36.68% (▼-0.75%) |
| LLaVA 1.5 13B | 40.12% | 38.78% (▼-1.34%) |
| CaD-VI 7B | 38.40% | 40.44% (▲+2.04%) |
| CaD-VI 13B | 40.16% | 43.09% (▲+2.93%) |

Table 3: Performance on SEED-Bench video partitions by feeding one or two frames into the LMMs.

| Model | SEED-Image |
|---|---|
| LLaVA 1.5 7B | 67.34% |
| CaD-VI 7B | 67.48% |
| LLaVA 1.5 13B | 68.83% |
| CaD-VI 13B | 69.11% |

Table 4: Performance on SEED-Bench image partitions for evaluation of general VL capabilities with single-image input.

Additionally, to verify that introducing multi-image CaD data into the tuning does not lead to catastrophic forgetting of general single-image input LMM capabilities, we also evaluate the SEED-Bench Image partitions and report the results in Table 4. Here we directly compare to same architecture baseline of LLaVA 1.5 fine-tuned using its single-image LLaVA mix 665K data. Table 4 demonstrates that our CaD tuning indeed preserves the competence in single-image understanding. Evaluation on more general VL benchmarks like MME (Fu et al., 2023) and MMBench (Liu et al., 2023c) can be found in the Appendix.

| | Training Data | BISON | SVO | EQBEN | COLA | CaD-QA |
|---|---|---|---|---|---|---|
| A: | LLaVA mix | 54.00% | 46.80% | 17.50% | 7.62% | 2.54 |
| B: | LLaVA mix + ScDiff orig. annot. | 92.67% | 90.07% | 22.50% | 33.81% | 2.90 |
| C: | LLaVA mix + ScDiff our annot. (from scratch) | 88.67% | 90.80% | 38.33% | 36.67% | 3.17 |
| D: | LLaVA mix + ScDiff our annot. (refined from orig. annot.) | 94.67% | 91.80% | 32.50% | 34.76% | 3.17 |
| E: | LLaVA mix + CaD-Inst$^{V1}$ | 92.00% | 92.27% | 34.17% | 36.67% | 3.27 |
| F: | LLaVA mix + CaD-Inst$^{V1}$ + ScDiff our annot. (refined from orig. annot.) | 95.33% | 92.73% | 39.17% | 40.95% | 3.29 |

Table 5: Ablation of phase-2 data collection from 71K image pairs in Scene-Difference (ScDiff). We use CaD-LLaVA$^{V1}$ to generate CaD on ScDiff either from scratch or by refining from the original annotation of unstructured difference-only summaries. Training settings in E and F lead to our CaD-LLaVA$^{V1}$ and CaD-LLaVA$^{V2}$ models correspondingly.

## 6 ABLATIONS

**Phase-2 Data Collection analysis.** Our Phase-2 data collection introduced in Sec. 3.3 can be used to leverage image pairs from various sources for producing effective CaD instructions. We first ablate the data collection from the 71K image pairs in Scene-Difference (Li et al., 2023a) (ScDiff) which contains annotation of unstructured difference-only summaries. As shown in Table 5, training with original annotation of difference-only summaries (row B) significantly improves on the baseline of training with LLaVA data only (row A). Then we show that using CaD-LLaVA$^{V1}$ to generate CaD instructions on ScDiff remarkably improves further, either if used from scratch (row C) or by refining from the original annotation (row D, also illustrated in Fig. 2 bottom row). Training with our re-annotation from scratch outperforms the original annotation on all datasets except for BISON. Our re-annotation by refining the original annotation leads to a more balanced performance

| | Training Data | BISON | SVO | EQBEN | COLA | CaD-QA |
|---|---|---|---|---|---|---|
| A: | LLaVA mix | 54.00% | 46.80% | 17.50% | 7.62% | 2.54 |
| B: | LLaVA mix + A/G orig. captions only | 55.33% | 55.67% | 3.33% | 2.86% | 2.78 |
| C: | LLaVA mix + A/G our annot. (from scratch) | 90.00% | 88.53% | 40.83% | 42.86% | 3.21 |
| D: | LLaVA mix + A/G our annot. (given orig. captions) | 88.00% | 86.87% | 43.33% | 30.48% | 3.06 |

Table 6: Ablation of phase-2 data collection from 66K pairs of video frames in Action Genome and GEBC (A/G). We use CaD-LLaVA$^{V1}$ to generate CaD on A/G either from scratch or with the prior information from the original frame captions.

| | Training Data | BISON | SVO | CaD-QA | VG comm. | VG diff. |
|---|---|---|---|---|---|---|
| A: | LLaVA mix | 54.00% | 46.80% | 2.54 | 1.79 | 1.75 |
| B: | LLaVA mix + t2i retriev. | 58.00% | 51.33% | 2.47 | 1.58 | 1.46 |
| C: | LLaVA mix + comm. | 64.67% | 79.73% | 3.23 | **2.67** | 2.52 |
| D: | LLaVA mix + diff. | 55.33% | 72.13% | 3.24 | 1.97 | 2.89 |
| E: | LLaVA mix + comm. + diff. | 72.00% | 82.60% | 3.24 | 2.13 | 3.42 |
| F: | LLaVA mix + comm. + diff. + t2i retriev. | 92.00% | 92.27% | 3.27 | 2.21 | 3.69 |
| G: | (F) + CaD-Inst$^{V2}$ | **95.33%** | **92.73%** | **3.29** | 2.32 | **3.85** |

Table 7: Ablation on components in the instruction data. Training settings in F and G lead to our CaD-LLaVA$^{V1}$ and CaD-LLaVA$^{V2}$ models correspondingly. Here *t2i retriev.* refers to the text-to-image retrieval task (see Sec. 3.1). Training settings in F and G lead to our CaD-LLaVA$^{V1}$ and CaD-LLaVA$^{V2}$ models correspondingly.

improvement and is used as the phase-2 instruction data CaD-Inst$^{V2}$ . We combine this with our phase-1 data CaD-Inst$^{V1}$ and demonstrate the further performance boost in row F of Table 5.

In order to show the robustness of CaD data collection capability using our CaD-LLaVA$^{V1}$ model, we also explore applying our phase-2 data collection to visually similar frames from user videos in Action Genome and GEBC (A/G). In Table 6, we first train a baseline using original frame captions only and a simple instruction task of image description (row B), which leads to a significant performance drop on EQBEN and COLA, and minimal improvement on other datasets. Then we use our CaD-LLaVA$^{V1}$ to generate CaD instructions on the frame pairs either from scratch (row C) or conditioned on the frame captions (row D). Interestingly, on most datasets CaD instructions generated by our CaD-LLaVA$^{V1}$ from scratch are found to be more effective than ones generated using original captions conditioning, likely due to lack of detail in these captions. This once again demonstrates that our model is effective in generating CaD instructions on unlabeled data. In the Appendix, we further show that our phase-2 data collection is effective on out-of-distribution video-surveillance data of Spot-the-diff (SpotDiff) dataset (Jhamtani & Berg-Kirkpatrick, 2018b).

**Analysis of CaD Instruction Data Components** We verify the effectiveness of the components in our instruction data by ablating on the different combinations of our tuning tasks, including: (i) commonality summary (*comm.*); (2) difference summary (*diff.*); and (iii) text-to-image retrieval (*t2i retriev.*) in Table 7. Training solely on the t2i retrieval task (row B) leads to minimum performance improvement on BISON and SVO Probes, and performance degradation on the three benchmarks of the open-ended tasks due to lacking of any CaD learning. Training with the commonality (row C) and difference summary (row D) tasks separately lead to a significant boost on the VG comm (2.67) and VG diff (2.89) tasks correspondingly. Training with combinations of the three tasks (F) boosts the performance in comparison to the case of each single component, except for VG comm where the commonality training (row C) leads to better results on this task. Finally, combining phase-1 and phase-2 data (row G) leads to further performance boosts on most of the benchmarks.

# 7 CONCLUSIONS, LIMITATIONS, AND BROADER IMPACT

We are contributing CaD-VI - an effective, two-phase strategy for collecting Commonalities and Differences (CaD ) Visual Instruction (VI) data, resulting in the also contributed large scale CaD-Inst with 349K samples for verified improvement of CaD and related image and text comparative capabilities of LMMs. Additionally, we contribute CaD-QA - a benchmark of 7.6K open-ended QA to directly evaluate CaD capabilities between pairs of images. We extensively evaluate and validate our CaD-VI approach, showing it leads to substantial improvements in CaD abilities and related tasks. We further show how the very few existing CaD resources are complementary to our approach and can be further refined automatically using our CaD-VI . We believe that our work contributes to the important investigation and improvement of (currently somewhat missing) CaD abilities of modern LMMs and leads to exciting future work of CaD VI tuning.

**Limitations** Currently, our CaD-VI only focuses on the CaD between two images, and we leave the extension of understanding CaD and group relations on three or more images to future work.

**Broader Impact** Our CaD-VI , CaD-Inst , and CaD-QA significantly contribute to the understanding and improvement of CaD capabilities in LMMs, and are intended to enhance the applicability and utility of AI across various fields, from robotics to industrial applications. However, this LMM improvement could also lead to job displacement, as these models could increasingly automate complex tasks traditionally performed by humans.

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

# A INTRODUCTION

In this appendix, we first provide our **source code** (Sec. B).

As additional results, we include more **evaluations on the open-ended CaD-QA** with different LLMs, in-context examples of scoring cases and human study in Sec. C.1. Then we report results of CaD-VI on **two more general vision-language benchmarks** (Sec. C.2). Further include the evaluation of a **video LMM** in Sec. C.3. We report the **error bars** (Sec.C.4), analyze the **Phase-2 data collection on Out-Of-Distribution data** (Sec. C.5). Finally, we show **qualitative results** of the collected CaD summaries (Sec. C.6), and compare LMM predictions on our CaD-QA benchmark (Sec. C.7), and LMM predictions on the BISON dataset (Sec. C.8).

For further insights into our approach CaD-VI , we report more **statistics on our generated data** (Sec. D.1), and **statistics on the external evaluation datasets** (Sec. D.2). We provide more **implementation details** (Sec. E) including the specifics of baseline methods, data generation, training and evaluation details.

At last, we provide the **list of assets** (Sec. F) used in this project.

# B SOURCE CODE

The source code is provided in the supplementary materials of `CaD-VI.zip`.

# C ADDITIONAL RESULTS

## C.1 ADDITIONAL EVALUATIONS OF OPEN-ENDED CaD QA

| Model | Mixtral 8×7B | LLaMA 3.1 70B | GPT4o mini |
|---|---|---|---|
| SparklesChat | 3.01 | 2.91 | 2.62 |
| Otter | 2.20 | 1.70 | 1.66 |
| MMICL | 2.01 | 1.97 | 2.00 |
| EMU2-Chat | 1.20 | 1.26 | 1.34 |
| InternLM-XComposer2-VL | 2.90 | 2.79 | 2.61 |
| LLaVA 1.6 7B | 3.10 | 2.80 | 2.54 |
| LLaVA 1.6 13B | 3.19 | 3.00 | 2.67 |
| LLaVA 1.5 7B | 2.54 | 1.98 | 1.86 |
| LLaVA 1.5 13B | 2.65 | 2.11 | 1.98 |
| CaD-VI 7B | 3.29 | 3.02 | 2.72 |
| CaD-VI 13B | **3.34** | **3.10** | **2.78** |

Table 8: Impact of different LLMs on the LLM-assisted evaluation of the open-ended CaD QA benchmark.

**Different LLMs.** In order to mitigate the bias from the same LLM used for evaluation and show the impact of different LLMs on the LLM-assisted evaluation, we further employ LLaMA 3.1 70B and GPT4o mini for the evaluation of CaD QA and report the resutls in Tab. 8. In case of LLaMA 3.1 70B and GPT4o mini, CaD-VI still outperforms all the other competitors. However, there is a drop in the margin of its outperformance in comparison to the case of Mixtral model assisted evaluation.

**Scoring standard descriptions.** We further explore the impact of scoring standard descriptions in the evaluation of open-ended CaD QA. We provide in-context examples for cases of different scores. In Tab. 9, we report the evauation results with and without in-context examples of scoring cases. In all cases. CaD-VI still outperforms the other competitors. Evaluation with in-context examples of ratings leads to drop of ratings on Mixtral 8×7B but slight increase of rating on LLaMA 3.1 70B. This could due to the better in-context learning capability of LLaMA 3.1.

**Human study.** Furthermore, we randomly sampled 150 open-ended questions from the evaluation benchmark and asked three volunteers to manually rate the predictions of the compared LMMs in the range between 0 and 5. To reduce the rating efforts, we include the 13B version of CaD-VI and LLaVA models in this task.

| Model
In-context | Mixtral 8×7B
No | Mixtral 8×7B
Yes | LLaMA 3.1 70B
No | LLaMA 3.1 70B
Yes |
|---|---|---|---|---|
| SparklesChat | 3.01 | 2.08 | 2.91 | 3.14 |
| Otter | 2.20 | 1.17 | 1.70 | 2.02 |
| MMICL | 2.01 | 1.72 | 1.97 | 2.40 |
| EMU2-Chat | 1.20 | 1.01 | 1.26 | 1.42 |
| InternLM-XComposer2-VL | 2.90 | 2.52 | 2.79 | 3.15 |
| LLaVA 1.6 7B | 3.10 | 2.06 | 2.80 | 2.97 |
| LLaVA 1.6 13B | 3.19 | 2.16 | 3.00 | 3.13 |
| LLaVA 1.5 7B | 2.54 | 1.56 | 1.98 | 2.18 |
| LLaVA 1.5 13B | 2.65 | 1.77 | 2.11 | 2.33 |
| CaD-VI 7B | 3.29 | 2.54 | 3.02 | 3.20 |
| CaD-VI 13B | **3.34** | **2.68** | **3.10** | **3.31** |

Table 9: Impact of in-context examples of scoring cases on the LLM-assisted evaluation of the open-ended CaD QA benchmark.

| Model | CaD-VI 13B | LLaVA 1.6 13B | LLaVA 1.5 13B | InternLM-XComposer2-VL | SparklesChat |
|---|---|---|---|---|---|
| Rating | 3.61 | 3.42 | 2.84 | 3.05 | 3.30 |

Table 10: Human evaluation on 150 randomly sampled questions from the open-ended CaD QA benchmark.

| Model | MMBench | MME Perception | MME Cognition |
|---|---|---|---|
| LLaVA 1.5 7B | 65.80% | 1498.09 | 274.64 |
| CaD-VI 7B | 65.38% | 1493.21 | 328.57 |
| LLaVA 1.5 13B | 69.07% | 1541.69 | 300.36 |
| CaD-VI 13B | 68.27% | 1530.61 | 306.07 |

Table 11: Evaluation of CaD-VI on general vision-language benchmarks MMBench and MME.

As shown in Tab. 10, the results indicate the human preference of answers from CaD-VI , which is aligned with the choice of LLMs. In the analysis of feedback from the human study, we also have some interesting conclusions: (1) The verbose descriptions with hallucinations from the talkative SparklesChat are better rated by humans than LLMs (2) InternLM-XComposer2-VL could generate correct and concise descriptions of visual contents but is not good at the task of comparison (3) LLaVA 1.6 could see more visual details than LLaVA 1.5 due to the AnyRes (any-resolution) pipeline which benefits the comparison reasoning. In this case, using an architecture with more visual tokens to focus on local regions of images would allow comparison of more visual details via the comparison visual instruction tuning.

## C.2 ADDITIONAL EVALUATIONS ON GENERAL VISION-LANGUAGE BENCHMARKS

In the main paper, we report performance of CaD-VI on the general vision-language benchmark SEED-Bench image (Tab. 2 in the main paper) and SEED-Bench video(Tab. 3 in the main paper), which verifies that introducing multi-image CaD data into tuning does not lead to catastrophic forgetting of general single-image input LMM capabilities.

Additionally, we compare the performance of CaD-VI to the original LLaVA models on MME (Fu et al., 2023) and MMBench (Liu et al., 2023c) in Tab. 11. We see that after introducing CaD data into tuning, there is only a slight performance drop of CaD-VI in comparison to the original LLaVA on MMBench and MME Perception. On MME Cognition tasks, CaD-VI even has some performance improvements.

## C.3 EVALUATION OF VIDEOLLAMA2

In the main manuscript, we include five models that train on samples with multiple input images, *i.e.* SparklesChat, Otter, MMICL, EMU2-Chat, InternLM-XComposer2-VL. We additionally report the performance of a recent video LMM VideoLLaMA2 (Cheng et al., 2024) on the benchmark

| Model | BISON | SVO | NLVR2 | EQBEN | COLA | CaD-QA |
|---|---|---|---|---|---|---|
| VideoLLaMA2 | 58.00% | 61.00% | 64.00% | 11.67% | 16.67% | 2.22 |
| CaD-VI 7B | 95.33% | 92.73% | 66.67% | 39.17% | 40.95% | 3.29 |
| CaD-VI 13B | 96.67% | 93.00% | 69.33% | 42.50% | 43.33% | 3.34 |

Table 12: Evaluation of VideoLLaMA2Cheng et al. (2024) on the benchmark datasets.

| Training Data | BISON | SVO | EQBEN | COLA | CaD-QA |
|---|---|---|---|---|---|
| LLaVA mix + CaD-LLaVA$^{V1}$ | 91.78% ± 1.02% | 92.33% ± 0.57% | 33.06% ± 0.96% | 34.64% ± 2.09% | 3.270 ± 0.002 |

Table 13: Average performance of the Phase-1 model CaD-LLaVA$^{V1}$ on multiple runs of training.

| | Training Data | BISON | SVO | Difference Spotting | CaD-QA |
|---|---|---|---|---|---|
| A: | LLaVA mix (L) | **54.00%** | 46.80% | 49.50% | 2.54 |
| B: | L + SpotDiff orig. annot. | 51.33% | 52.27% | 60.48% | 2.51 |
| C: | L + SpotDiff our annot. (refined from orig. annot.) | **54.00%** | **54.87%** | **66.67%** | **2.86** |

Table 14: Ablation of phase-2 data collection from 15K pairs of video frames in Spot-the-diff (Spot-Diff). We use CaD-LLaVA$^{V1}$ to generate CaD on SpotDiff by refining from the original human-annotated difference descriptions.

datasets. As shown in Tab. 12, our CaD-VI could outperform VideoLLaMA2 on all the benchmarks. The reason that the video LMM does not perform well on benchmarks of CaD capabilities could be that it is trained to understand a video as a spatio-temporal entity instead of multiple individual images.

## C.4 ERROR BARS

We run the training of the Phase-1 model CaD-LLaVA$^{V1}$ multiple times and report the average performance with standard deviation in Table 13. In most evaluation cases, the standard deviation is within around 1%.

## C.5 ABLATION ON PHASE-2 DATA COLLECTION - OOD CAD REFINEMENT

In Section 6 (main paper), we perform ablation the Phase-2 data collection. Here we further explore applying our phase-2 data collection on out-of-distribution (OOD) data of Spot-the-diff (SpotD-iff) dataset. The dataset contains distant-view frame pairs with very subtle changes from video-surveillance footage, which are OOD from most LMM training data.

In Table 14, we train with SpotDiff original human-annotated difference description (row B) and with our CaD-LLaVA$^{V1}$ generated CaD summaries which is refined from the original annotation (row C). We also evaluate on the Difference-Spotting partition on SEED-Bench 2 (Li et al., 2023c) which contains multi-choice questions based on frame pairs from SpotDiff. In data collection and training for this experiment, we only used the 15K training image pairs from SpotDiff which are not included in the Difference-Spotting SEED partition. The results in Table 14 verify that our phase-2 data collection using CaD-LLaVA$^{V1}$ is also effective on OOD data.

## C.6 QUALITATIVE RESULTS OF CAD SUMMARIES

In Fig. 2 (main paper), we illustrate the pipeline of our two-phase CaD-VI together with two examples of Phase-1 LLM-collected CaD summary and Phase-2 LMM-collected CaD summary. Here in Fig. 5, we provide two additional examples. Note that in Fig. 5(a), we only pass the captions with the instruction prompt (in Fig. 15) into the LLM. In Fig. 5(b), we pass the original annotation and both images with the instruction prompt (in Fig. 16) into the Phase-1 model. In the main paper (Table 5), we demonstrate the generated CaD summary without using the original annotation also leads to effective results.

Image 1        Image 2

(a)

Caption1: In this image there is a dining table on which there is a flower vase. There is a Champagne bottle. There is laptop. There is a headphone. There at the left side of the image there is a chair on which there is a bag. At the background of the image there is a wall and a window.

Caption2: In the image in the center we can see four persons were standing and they were smiling,which we can see on there faces. In front of them,there is a table. On table,we can see different types of books. In the center we can two persons were standing and the left lady is holding paper bag. In the background there is a brick wall,banner,belts,hanger,photo frames,toys,sign board,clothes,table and few other objects.

**LLM** : **Commonalities:**
Both images include indoor scenes with man-made objects. There is a table in both images, and the tables are surrounded by other objects.

**Differences:**
1. Object types: Image 1 contains a dining table, flower vase, Champagne bottle, laptop, headphone, chair, bag, wall, and window. Image 2 contains four people, a table, books, paper bag, brick wall, banner, belts, hangers, photo frames, toys, sign board, clothes, and other objects.
2. Counting the objects: There are 1 dining table, 1 flower vase, 1 Champagne bottle, 1 laptop, 1 headphone, 1 chair, 1 bag, 1 wall, and 1 window in Image 1. In Image 2, there are 4 people, 1 table, many books, 1 paper bag, 1 brick wall, 1 banner, several belts, hangers, photo frames, toys, 1 sign board, and multiple clothes.
3. Object actions: There are no specific actions involving objects in Image 1. In Image 2, four people are standing and smiling.
4. Object locations: In Image 1, the objects are placed on the dining table and the floor. In Image 2, the people are standing in front of the table, and the books are on the table.
5. Relative positions between objects: In Image 1, the chair is located to the left of the table. In Image 2, the paper bag is being held by one of the women, and the other objects are placed on the table or hung on the wall.

Image 1        Image 2

(b) 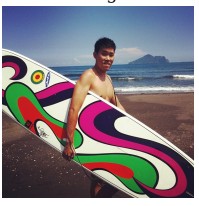 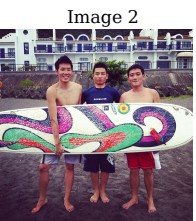

Original annotation: "*The surfboard in the first image is a very colorful one, while the surfboard in the second image is a decorated one. In the first image, there is only one person holding the surfboard, while in the second image, there are three people holding the surfboard.*"

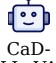 **CaD-LLaVA** : **Commonalities:**
Both images contain a surfboard as the main object. The surfboard is being held by people in both images. The images also have a natural background, with the first image showing water, a mountain, and the sky, and the second image showing a building, trees, and a fence.

**Differences:**
1. The number of people holding the surfboard is different. In the first image, there is only one person, while in the second image, there are three people.
2. The background in the first image is more natural, with water, a mountain, and the sky, while the background in the second image is more urban, with a building, trees, and a fence.
3. The first image has a more open and spacious setting, while the second image has a more crowded and enclosed setting.
4. The first image has a more dynamic and active feel, with the person holding the surfboard, while the second image has a more static and posed feel, with the three people standing

Figure 5: Examples of (a) Phase-1 LLM-collected CaD summary and (b) Phase-2 LMM-collected CaD summary

## C.7 QUALITATIVE RESULTS ON CAD-QA

In Fig. 6, Fig. 7 and Fig. 8, we show examples of Q&A pairs in our CaD-QA , together with the predicted answers from CaD-LLaVA$^{V2}$ model and the vanilla LLaVA 1.5 model. We also report the LLM ratings for the predicted answers. The vanilla LLaVA model has incorrect answers by either mistakenly combining the contents in two images (Fig. 6(b), *the man is standing in front of the toilet while holding an umbrella*), omitting one of the images (Fig. 7(a), Fig. 8(a)), attending to the incorrect image (Fig. 7(c)) or hallucinating non-existent contents (Fig. 8(b)). The failure demonstrates the lacking of capability of properly comparing two images. At the same time, our CaD-Inst$^{V2}$ manages to correctly differentiate between the two images, attend to the corresponding content asked in the question and draw a summary of comparison.

## C.8 QUALITATIVE RESULTS ON BISON

In Fig. 9, we illustrate some examples of the binary image selection task on BISON. We instruct the LMMs to give both the selection answer and also the reasoning for the selection. Here we compare the vanilla LLaVA 1.5 and our CaD-LLaVA$^{V2}$ . The LLaVA model, even if it captures the relevant content in some cases, has confusion differentiating the two images (Fig. 9(a)(b)). For our CaD-LLaVA$^{V2}$ , the key reasoning that leads to the correct answer is always covered in the structured difference summary.

Image 1          Image 2

(a) 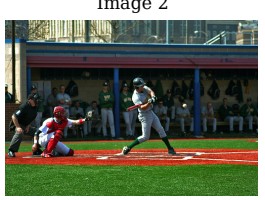

Q: What type of outdoor recreational activities are depicted in the two images?

GT: In Image 1, a group of young people are engaged in a game of frisbee in a park, while in Image 2, an ongoing baseball game is shown, with players in uniforms and specialized equipment.

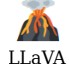 LLaVA : The two images depict a baseball game in progress, with a batter swinging a baseball bat and a catcher wearing a baseball glove. The scene takes place on a baseball field, where multiple people are participating in the game. The presence of a baseball glove and the batter's swinging motion indicate that the game is in action, and the players are actively engaged in the sport. ongoing baseball game is shown, with players in uniforms and specialized equipment.

LLM rating: 3

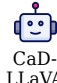 CaD-LLaVA : The two images depict different outdoor recreational activities. Image 1 shows a group of people playing with a frisbee on a grassy field, while Image 2 features a baseball game with a batter, a catcher, and an umpire. The activities in the images are distinct, with one involving a casual, informal game of frisbee, and the other being a more structured and organized baseball game.

LLM rating: 5

Image 1          Image 2

(b) 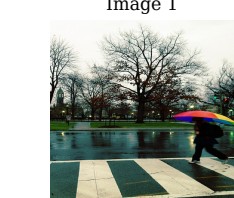 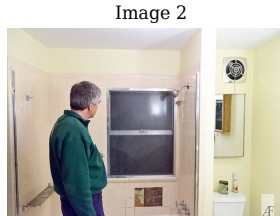

Q: What objects are present in Image 2 that are not in Image 1?

GT: In Image 2, there is a well-equipped shower, a bathtub, a commode, a mirror, a wooden shelf, and various items such as a green candle, air freshener spray, and silver faucet. These objects are not present in Image 1.

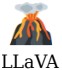 LLaVA : In Image 2, there is a toilet visible in the bathroom, which is not present in Image 1. The man is standing in front of the toilet while holding an umbrella.

LLM rating: 2

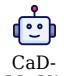 CaD-LLaVA : In Image 2, there are several objects that are not present in Image 1, including a man, a mirror, a sink, a toilet, a shower, a window, a door, and a ceiling. The man is standing in the bathroom, and the mirror, sink, toilet, and shower are all located in the same room. The window and door are also present in the bathroom, and the ceiling is visible at the top of the image.

LLM rating: 3

Image 1          Image 2

(c) 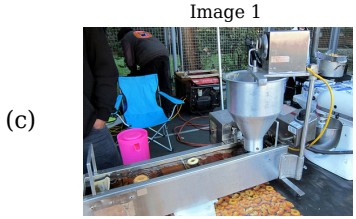 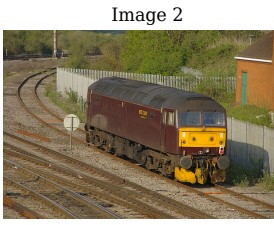

Q: Are there any humans visible in Image 2?

GT: No, there are no visible humans in Image 2, but the presence of a train driver is implied.

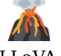 LLaVA : Yes, there is a man visible in Image 2, standing next to the train.

LLM rating: 0

CaD-LLaVA : No, there are no humans visible in Image 2.

LLM rating: 4

Figure 6: Examples of Q&A pairs in CaD-QA together with LMM predicted answers and the corresponding LLM evaluation rating for the prediction (Red and green texts denote incorrect and correct description).

Image 1           Image 2

(a)
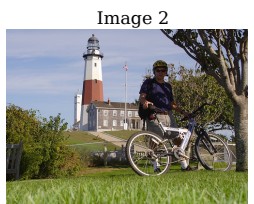

Q: What is the central figure in each image?

GT: In Image 1, the central figures are several horses and their riders, engaged in a polo game. In Image 2, the central figure is a man holding a mountain bike.

🌋 : The central figure in each image is a man riding a bicycle.

LLaVA    LLM rating: 2

🤖 : In Image 1, the central figure is a man riding a horse. In Image 2, the central figure is a man riding a bicycle.

CaD-LLaVA    LLM rating: 4

Image 1           Image 2

(b)
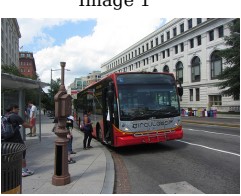 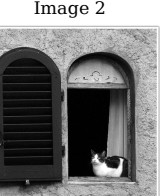

Q: What kind of mood do the two images convey?

GT: Image 1 conveys a sense of dynamism and energy in a modern urban setting. Image 2 evokes a feeling of tranquility and curiosity in a traditional, mysterious setting.

🌋 : The two images convey a calm and peaceful mood. In the first image, a cat is sitting in a window, looking out, which suggests a sense of relaxation and contentment. The second image shows a bus parked on the side of the road, with people waiting at a bus stop. The presence of the bus and the people waiting for it indicate a sense of order and routine, which can also contribute to a calm atmosphere. The combination of these two images creates a serene and tranquil scene, reflecting the everyday life of people and animals in a quiet, urban environment.

LLaVA    LLM rating: 2

🤖 : The two images convey different moods. Image 1 has a lively and active mood, with a bus, people, a pole, a dustbin, trees, buildings, and a cloudy sky. The presence of a bus and people suggests movement and activity. Image 2, on the other hand, has a calm and serene mood, with a cat sitting on a window sill and a wall in the background. The cat's presence and the stillness of the scene create a sense of tranquility.

CaD-LLaVA    LLM rating: 4

Image 1           Image 2

(c)
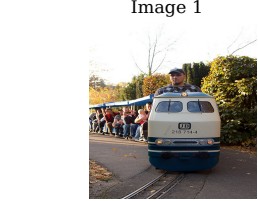 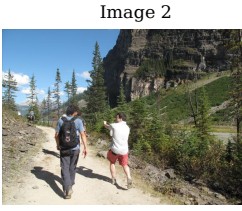

Q: What is the atmosphere like in Image 1?

GT: The atmosphere in Image 1 is lively and conveys a sense of joy and adventure for the passengers aboard the train.

🌋 : The atmosphere in Image 1 is lively and social, with a group of people walking down a dirt road or path, enjoying each other's company.

LLaVA    LLM rating: 3

🤖 : The atmosphere in Image 1 is lively and engaging, as there are people sitting on a train and taking pictures of the scenery. The presence of a train and the outdoor setting suggest a sense of adventure and excitement, as the passengers are enjoying their journey and capturing memories of the beautiful surroundings.

CaD-LLaVA    LLM rating: 5

Figure 7: Examples of Q&A pairs in CaD-QA together with LMM predicted answers and the corresponding LLM evaluation rating for the prediction (Red and green texts denote incorrect and correct description).

Image 1                Image 2

(a)

Q: Are the settings in the two images similar or different?

GT: The settings in the two images are different. Image 1 is set in a train station during winter, while Image 2 is at a beach during a sunny day.

: The settings in the two images are similar, as both depict men sitting on a bench or a wooden chair near a beach.

LLaVA    LLM rating: 0

: The settings in the two images are different. Image 1 features a train station with a train on the tracks, people walking on the platform, and buildings in the background. Image 2, on the other hand, shows two
CaD-    men sitting on a bench under a colorful umbrella, with a body of water and boats in the background.
LLaVA
LLM rating: 5

Image 1                Image 2

(b)

Q: Are there any plants or natural elements in the images?

GT: No, neither Image 1 nor Image 2 contains any plants or natural elements.

: Yes, there is a plant in the image, which is placed on the table.

LLaVA    LLM rating: 0

: No, there are no plants or natural elements in the images. Both images are indoor scenes with man-made
CaD-    objects.
LLaVA    LLM rating: 5

Image 1                Image 2

(c)

Q: What objects are present in each image?

GT: Image 1 contains a wooden and metallic bench, a cellphone, and some graffiti, while Image 2 features a blue metal plow, horses, and rocky soil.

: In the first image, there is a man and a horse, while in the second image, there are two horses and a man.

LLaVA    LLM rating: 1

: In Image 1, there is a woman sitting on a bench, and in Image 2, there are two horses, a cart, and three people.
CaD-
LLaVA    LLM rating: 2

Figure 8: Examples of Q&A pairs in CaD-QA together with LMM predicted answers and the corresponding LLM evaluation rating for the prediction (Red and green texts denote incorrect and correct description).

Image 1          Image 2

(a)

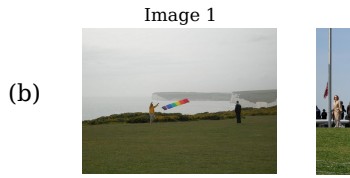

Text query: A large desk has a bookshelf above it.

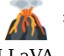
LLaVA
: Image 1.
The image with the bookshelf above the desk is more suitable for the caption "A large desk has a bookshelf above it." because it clearly shows a bookshelf above the desk, which is not visible in the other image. In the first image, the bookshelf is located above the desk, while in the second image, the bookshelf is not visible. The first image also has a laptop and a cell phone on the desk, which are not present in the second image

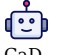
CaD-LLaVA
: Image 2.
Reasons:
* Image 2 contains a bookshelf with books and other items, while Image 1 does not have a bookshelf.
* Image 2 has a desk with a laptop, monitor, and other items, while Image 1 has a desk with a laptop, monitor, and other items.
* Image 2 has a wall with posters and a board, while Image 1 does not have a wall with posters or a board.
* Image 2 has a more cluttered and detailed scene compared to Image 1.

Image 1          Image 2

(b)

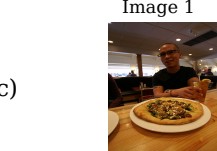

Text query: A group of men preparing to fly a kite being filmed by another man.

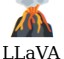
LLaVA
: Image 1.
The image with the man holding a camera captures the essence of the scene better. In this image, there is a man holding a camera, and another man is holding a kite. The presence of the camera indicates that the scene is being filmed, which is the main focus of the image. In contrast, the image with the man holding a camera and two other men standing in the grass does not clearly convey the act of filming. The presence of the kite in the image also adds to the visual interest and context of the scene.

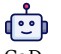
CaD-LLaVA
: Image 2.
Reasons:
1. The number of people: Image 1 has two people, while Image 2 has four people.
2. Actions: In Image 1, one person is flying a kite, and the other is standing. In Image 2, all three people are standing, and one person is holding a camera.

Image 1          Image 2

(c)

Text query: A man in a white sweater sitting at a table eating a whole pizza pie.

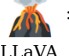
LLaVA
: Image 1.
The image with the man in a white sweater sitting at a table eating a whole pizza pie is more suitable for the caption because it clearly shows the man eating a pizza and wearing a white sweater. The other image, B, shows a man cutting a pizza with a knife and fork, but it does not provide enough context to determine if he is wearing a white sweater or not. The image A provides a more accurate representation of the man eating a pizza and wearing a white sweater, which is the main focus of the caption.

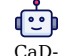
CaD-LLaVA
: Image 2.
Reasons:
1. The man in Image 2 is sitting at a table and eating a whole pizza pie, while the man in Image 1 is sitting at a table and holding a glass of beer.
2. The man in Image 2 is wearing a white sweater, while the man in Image 1 is not wearing a sweater.
3. The man in Image 2 is cutting the pizza with a knife, while the man in Image 1 is not cutting anything.

Figure 9: Examples of predictions of the binary image selection task on BISON (red and green texts denote incorrect and correct predictions). We instruct the LMMs to, besides the selection, also give a reasoning for the answer.

# D  DATASET STATISTICS

## D.1  GENERATED DATA STATISTICS

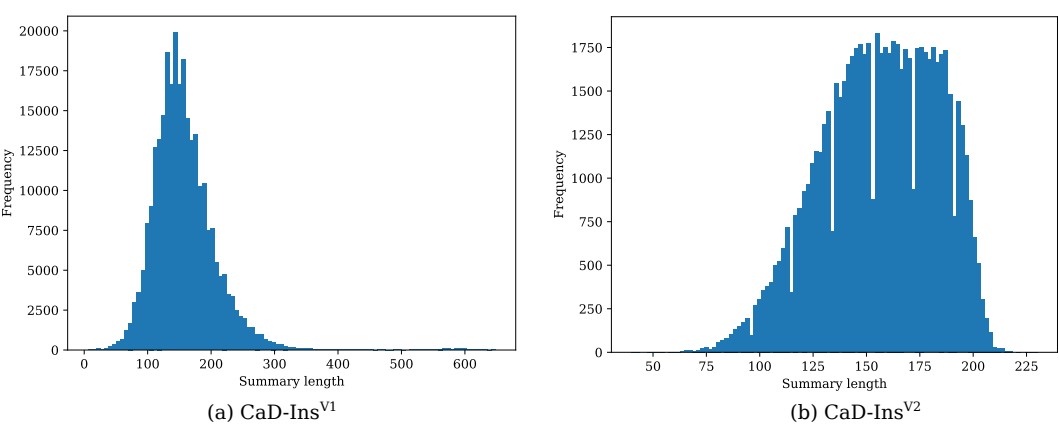

(a) CaD-Ins$^{V1}$

(b) CaD-Ins$^{V2}$

Figure 10: Distribution of length of CaD summaries (in terms of number of words) in (a) CaD-Inst$^{V1}$ and (b) CaD-Inst$^{V2}$

**CaD-Inst$^{V1}$ and CaD-Inst$^{V2}$ .** In CaD-Inst$^{V1}$ , we collected structured summaries of CaD for 278K image pairs, with an average length of 157 words (40 for commonalities and 117 for differences). In CaD-Inst$^{V2}$ , we collected summaries of CaD for 71K images pairs used in Scene-Difference (Li et al., 2023a), with an average length of 156 words (28 for commonalities and 128 for differences). We demonstrate the distribution of CaD summary length (number of words) in CaD-Inst$^{V1}$ (Fig. 10(a)) and in CaD-Inst$^{V2}$ (Fig. 10(b)).

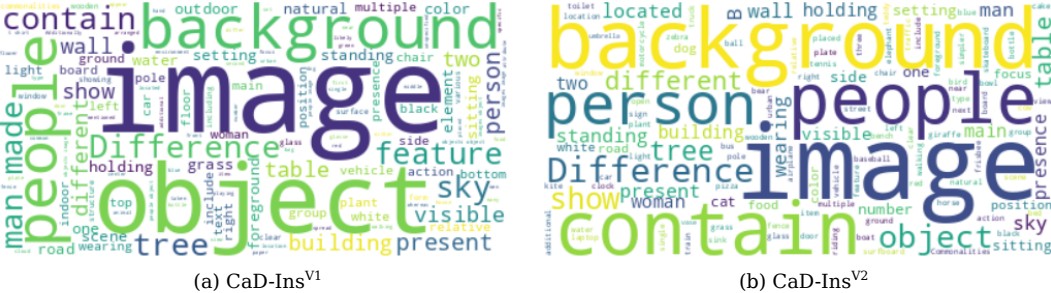

(a) CaD-Ins$^{V1}$

(b) CaD-Ins$^{V2}$

Figure 11: Word clouds of CaD summaries in (a) CaD-Inst$^{V1}$ and (b) CaD-Inst$^{V2}$

In Fig. 11, we also illustrate the cloud of words covered in the CaD summaries in CaD-Inst$^{V1}$ (Fig. 11(a)) and in CaD-Inst$^{V2}$ (Fig. 11(b)).

In the main paper, we mentioned that the collected summaries are structured according to approximate 6 axes of characteristics: *object types, attributes, counting, actions, locations* and *relative positions*. Note that the characteristics appear unevenly on a case-to-case basis based on the LLM decision on individual samples. In Fig. 3(a)(main paper), we illustrate the distribution of these sample-specific characteristics in a Sunburst chart. Here in Fig. 12, we also illustrate the distribution of these characteristics (*e.g.* object types, action of people, surrounding environments, *etc.*) in CaD summaries in the Phase-1 data collection CaD-Inst$^{V1}$ . The structured differences are summarized in terms of these characteristics (see Fig. 5(a) for an example of structured difference summary in terms of several characteristics). The visual instruction tuning guides the model to compare images in terms of these detailed characteristics.

In the main paper, we introduced that we collect 278K image pairs with different levels of similarity between their captions. We measure the similarity between two captions by counting the number of overlapping nouns in the corresponding captions. Here we show the distribution of the number of

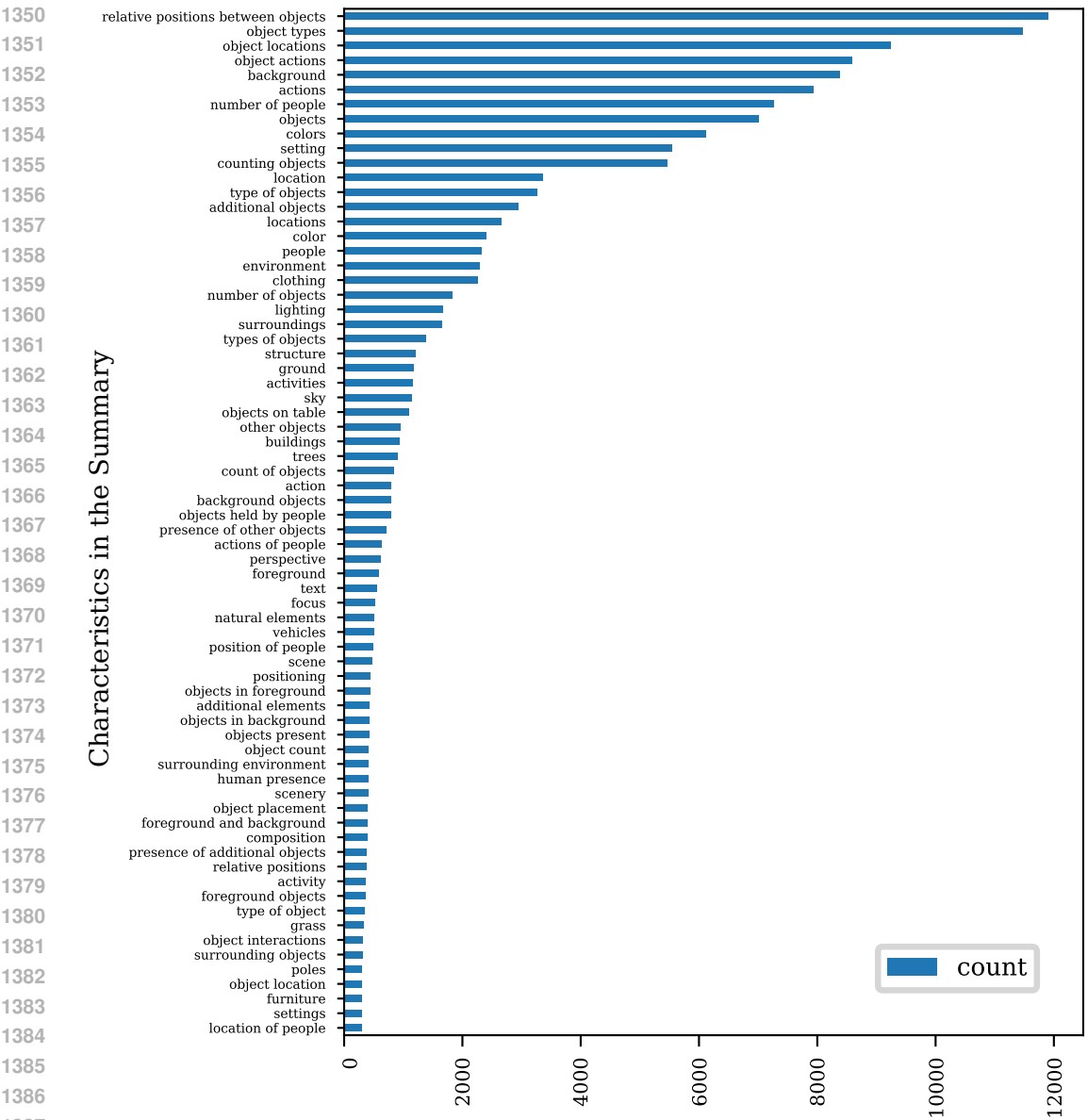

Figure 12: Distribution of sample-specific characteristics (*e.g.* object types, action of people, surrounding environments, *etc.*) in CaD summaries in CaD-Inst$^{V1}$ . The distribution of these sample-specific characteristics is also shown in a Sunburst chart in Fig. 3(a)(main paper).

overlapping nouns in Fig. 13(a). We see that we cover image pairs with different levels of caption-caption similarity. Furthermore, we use the CLIP ViT-B/32 model (Radford et al., 2021) to compute the similarity scores between the two images in each pair and report the distribution in Fig. 13(b). We verify that image pairs of diverse similarity levels are covered in our Phase-1 data collection CaD-Inst$^{V1}$ .

**CaD-QA .** Our CaD-QA benchmark contains 7.5K open-ended questions with answers. Here we show the distribution of questions types (first 5 words) and answer types (first 3 words) in Sunburst charts in Fig. 14. There are diverse question categories covered such as *Yes/No* questions, *What* questions on scene characteristics such as objects, attributes and setting, and also requests to describe specific characteristics in details.

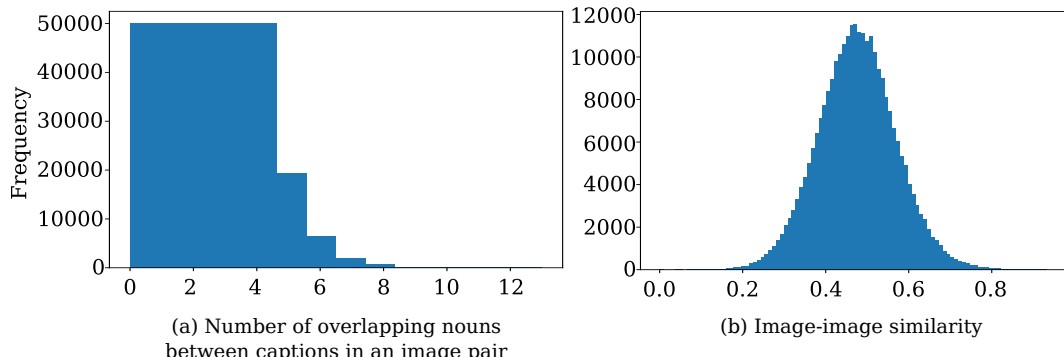

(a) Number of overlapping nouns
between captions in an image pair

(b) Image-image similarity

Figure 13: Distribution of (a) number of overlapping nouns between captions in an image pair and (b) image-image similarities in the 278K image pairs in CaD-Inst$^{V1}$

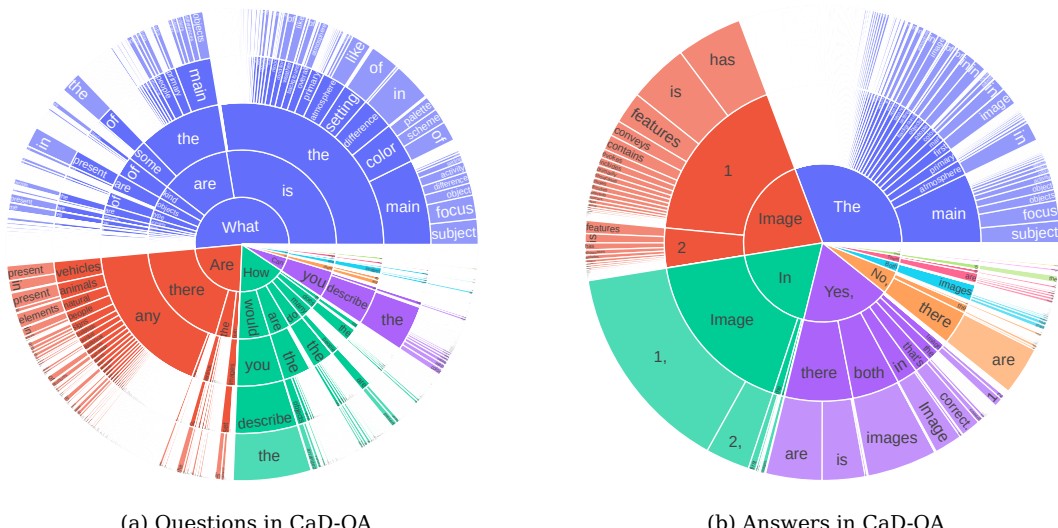

(a) Questions in CaD-QA

(b) Answers in CaD-QA

Figure 14: Distribution of (a) questions (first 5 words) and (b) answers (first 3 words) in the evaluation benchmark CaD-QA .

## D.2 STATISTICS OF EXTERNAL EVALUATION DATASETS

We evaluate on several external VQA benchmarks of closed-ended and open-ended questions. Here we give a brief introduction on the contents and statistics.

**BISON** is a dataset for the binary image selection task (Hu et al., 2019). There are 150 samples in the evaluation benchmark, each sample consisting of a pair of two visually similar images and a query caption. Only one image correctly matches with the query caption. It measures the ability of the LMMs to relate fine-grained text content in the caption to visual content in the images.

**SVO Probes** is a benchmark designed to probe for subject, verb and object understanding in vision-language models (Hendricks & Nematzadeh, 2021). In the benchmark, each sample consists of a pair of two images and a query sentence, where only one image correctly matches with the query sentence. The negative image differs from the positive image with regard to either the subject, the verb or the object. There are 36.8K samples in the dataset. For efficient evaluation, we randomly select 1500 samples that can be divided into 3 partitions *subject*, *verb* and *object* where each partition has 500 samples with the image pair contradiction in either subject, verb or object.

**EQBEN** is a benchmark that focuses on visual minimal change between two images (Wang et al., 2023). Each sample in the benchmark consists of a pair of two images with subtle visual changes and two corresponding captions. The dataset is comprised of frames from natural video datasets such as

YouCook2 (Zhou et al., 2018), Action Genome (Ji et al., 2020) and GEBC (Wang et al., 2022), as well as sythetic image pairs with subtle differences generated by the photo-realistic scene generator Kubric (Greff et al., 2022) and the diffusion model Stable-Diffusion (Rombach et al., 2022). We employ an EQBEN subset[1] which is released by the authors in (Wang et al., 2023) for evaluating the performance of LMMs specifically. The subset consists of 120 samples, comprised of frame pairs from Action Genome and GEBC, image pairs with changes in attributes, count and location generated by Kubric, and image pairs with style change generated by Stable-Diffusion. For each sample, we perform the binary image selection task twice, feeding one of the descriptions for image selection at a time. The sample is considered positively answered only when both selection tasks are correctly solved.

**COLA** is a benchmark for evaluating the capabilities of vision-language models on representing simple compositions by combing objects with their attributes (Ray et al., 2023). Each sample in the benchmark consists of two images with two corresponding captions. The two images have attributes and objects that are swapped in the captions, *e.g. large tree to the right of little short green tree*, and *tall green tree to the right of large tall green tree*. We employ the partition of *multi-object setting* in the benchmark which consists of 210 image pairs and captions. Similar to evaluation on EQBEN, we perform the binary image selection task twice for each sample.

**NLVR2** is a benchmark for evaluation of the visual reasoning with natural language task which aesses the ability of LMMs to predict whether a sentence is true about a pair of images (Suhr et al., 2019). The task focuses on understanding of compositionalities in terms of relations, comparisons and counting. We use the subset of 150 samples provided in SparklesChat (Huang et al., 2023) for a fair comparison.

**SEED-Bench** is an evaluation benchmark on comprehensive vision-language understanding, consisting of 19K multiple choice questions (Li et al., 2023d). The are two major categories in the benchmark: *SEED-Image* with 14K samples and *SEED-Video* with 5K samples. SEED-Image consists of 9 dimensions: scene understanding, instance identity, instance attributes, instance location, instance counting, spatial relation, visual reasoning and text understanding. All samples contain only a single input image. SEED-Video consists of 3 dimensions: action recognition, action prediction and procedure understanding. The videos are from Something-Something-v2 (Goyal et al., 2017a), EPIC-Kitchen (Damen et al., 2022) and Breakfast (Kuehne et al., 2014).

## E IMPLEMENTATION DETAILS

### E.1 BASELINES

**SparklesChat (Huang et al., 2023)** is finetuned from the first-stage pretrained model of MiniGPT4 (Zhu et al., 2023a). The model is finetuned with their collected multi-image dialogue data. SparklesChat follows the architecture of MiniGPT4 and uses Vicuna 7B (Chiang et al., 2023), EVA-CLIP ViT-G/14 (Fang et al., 2023) with a Q-Former from BLIP-2 (Li et al., 2023e). We use the model weights and instruction templates available at `https://github.com/HYPJUDY/Sparkles`.

**Otter (Li et al., 2023b)** is finetuned from the OpenFlamingo model (Awadalla et al., 2023) with the collected multimodal in-context instruction-response data in MIMIC-IT (Li et al., 2023a). We use their most recent open-sourced version Otter-Image-LLaMA7B-LA-InContext available at `https://huggingface.co/luodian/OTTER-Image-LLaMA7B-LA-InContext`.

**MMICL (Zhao et al., 2024)** is based on the InstructBLIP model (Dai et al., 2023). The model is finetuned their own collected multimodal in-context learning datast consisting of interleaved text-image inputs, inter-related multiple image inputs and multimodal in-context learning inputs. We evaluate with their model of the largest scale MMICL-InstructBLIP-T5-XXL, available at `https://huggingface.co/BleachNick/MMICL-Instructblip-T5-xxl`.

---

[1]`https://entuedu-my.sharepoint.com/:u:/g/personal/tan317_e_ntu_edu_sg/ETkpKSsmun1MpBw7FqfUUS8BwTX2gKkTQkDFsfOGCw-9yA?e=KGtpg0`

**EMU2-Chat (Sun et al., 2024)** is a generative multimodal model trained on large-scale multimodal sequences. The model consists of pretrained EVA-02-CLIP-E-plus (Sun et al., 2023b) and LLaMA-33B (Touvron et al., 2023a). The model weights and inference code are available at `https://huggingface.co/BAAI/Emu2-Chat`.

**InternLM-XComposer2-VL (Zhang et al., 2023a)** consists of CLIP ViT-L (Radford et al., 2021) and InternLM2-7B (Team, 2023). The model weights of the InternLM-XComposer2-VL-7B and inference code are available at `https://huggingface.co/internlm/internlm-xcomposer2-vl-7b`.

**LLaVA 1.5 (Liu et al., 2023a)** is an improved version from LLaVA (Liu et al., 2023b) with CLIP-ViT-L-336px (Radford et al., 2021) as the visual backbone and Vicuna 1.5 (Zheng et al., 2023) as the LLM. Our visual instruction tuning is performed using the open-sourced code of LLaVA 1.5. We train on the first-stage pretrained weights of LLaVA 1.5 via LoRA finetuning. We evaluate both LLaVA 1.5 7B lora and LLaVA 1.5 13B lora as baselines. The models are available at `https://huggingface.co/liuhaotian/llava-v1.5-7b-lora` and `https://huggingface.co/liuhaotian/llava-v1.5-13b-lora`.

**LLaVA 1.6 (Liu et al., 2024)** is an improved version from LLaVA 1.5 with increased input image resolution and improved mixture of instruction tuning data. The 7B and 13B versions are avaible on Huggingface at `https://huggingface.co/liuhaotian/llava-v1.6-vicuna-7b` and `https://huggingface.co/liuhaotian/llava-v1.6-vicuna-13b`. However, the training code is not yet available.

### E.2 IMPLEMENTATION DETAILS

---

**System prompt:**
You are an AI visual assistant and you are seeing two images. The two images are provided with two captions, each describing the content of an image. Your task is to summarize the commonalities and differences between the two images. Answer as you are seeing the images. Summarize the commonalities and differences about the visual content of the two images, including the object types, object attributes, counting the objects, object actions, object locations, relative positions between objects, etc.

**User prompt:**
Please summarize the commonalities and differences between the following two images:
Image 1:<caption1>
Image 2:<caption2>
Commonalities:

---

Figure 15: Prompt for the task of Phase-1 LLM-based CaD summary.

**Data Collection.** In Phase-1, we leverage the Mixtral 8x7B Instruct v0.1 model[2] with 8-bit inference for data generation. We set the batch size to 16 and max new token to 750. The prompt for the task of LLM-based CaD summary is given in Fig. 15. The generation with batch 16 fits to an A100 80G GPU.

In Phase-2, we leverage the Phase-1 model CaD-LLaVA$^{V1}$ 13B model to generate CaD summary on additional image pairs. The temperature, max new tokens and number of beams are set to 0, 256 and 1. The prompt for the task of LMM-based CaD summary is given in Fig. 16.

For collecting open-ended QAs in CaD-QA , we first use the LMM to generate the CaD summaries based on the image captions (see Fig. 15). Then we prompt the LLM with both the image captions and the CaD summary, instructing it to generate a multi-turn conversation with several rounds of Q&A. We also provide some in-context samples to demonstrate the desired layout. The prompt for the task of generating Q&A pairs based on both image captions and the CaD summary is illustrated in Fig. 17.

**Training.** We perform visual instruction tuning following the configuration in LLaVA 1.5. We set the batch size to 128 and train for one epoch. The learning rate for LLM with LoRA and for the

---

[2] Huggingface source: `https://huggingface.co/mistralai/Mixtral-8x7B-Instruct-v0.1`

---

**System prompt:**
A chat between a curious user and an artificial intelligence assistant. The assistant gives helpful, detailed, and polite answers to the user's questions.

**User prompt:**
Image 1: <image>
Image 2: <image>
Here are some context of the difference between the two images:
<description>
Based on the two images and the context, summarize the commonalities and differences about the visual content of the two images, including the object types, object attributes, counting the objects, object actions, object locations, relative positions between objects, etc.

---

Figure 16: Prompt for the task of Phase-2 LMM-based CaD summary.

projector are set to $1 \times 10^{-4}$ and $2 \times 10^{-5}$ correspondingly. The LoRA rank and alpha values are set to 128 and 256. The training experiments are run on 4×A100 80G GPUs.

**Inference.** For VQA inference, the temperature, max new tokens and number of beams are set to 0, 256 and 1.

**LLM-assisted Evaluation** We leverage the Mixtral 8×7B model for LLM-assisted evaluation on open-ended questions. We feed the question, correct answer and the predicted answer into the LLM and instruct it to provide a rating between 0 and 5. The prompt for generating the evaluation rating is given in Fig. 18.

**System prompt:**
You are an AI visual assistant, and you are seeing two images. The two images are provided with two captions, each describing the content of an image. Additionally, you are provided with a summary of the commonalities and differences between the two images. Design a conversation between you and a person asking about the two images based on the commonalities and differences between the two images. The answers should be in a tone that a visual AI assistant is seeing the two images and answering the question. Ask diverse questions and give corresponding answers. Include questions about the visual content of the two images, including the object types, object attributes, counting the objects, object actions, object locations, relative positions between objects, etc.

**User prompt:**
Please design a conversation with several rounds of questions and answers, where a person asks about the two images based on the commonalities and differences between the two images, and you answer.

Image 1: In this picture we can see two women are taking picture, side there is a glass and bottle are placed on the table, behind we can see some people are sitting on the chairs.
Image 2: In this image, we can see some buildings. There is a bridge on the right side of the image. There is a person and bus in the middle of the image. There is a wall beside the road. There are divider cones at the bottom of the image. At the top of the image, we can see the sky.
Summary: Commonalities: The two images have the presence of people and man-made structures. In both images, the people are engaged in different activities - taking a picture in Image 1 and standing in Image 2. The man-made structures include a table with objects in Image 1 and buildings, a bridge, a wall, and divider cones in Image 2.
Differences:
1. Scene: Image 1 is set in an indoor or semi-enclosed space, such as a café or restaurant, while Image 2 is set outdoors, in a cityscape or urban environment.
2. Objects: Image 1 contains objects like a glass, bottle, and chairs, which are not present in Image 2. On the other hand, Image 2 has elements like a bridge, wall, and divider cones, which are not in Image 1.
3. People: In Image 1, there are two women taking a picture, while in Image 2, there is a single person standing near a bus.
4. Actions: The people in Image 1 are taking a picture, while the person in Image 2 is not engaged in any specific action.
5. Locations: The scene in Image 1 is more confined, with a table and chairs, while Image 2 has a wider view, including a road, buildings, and the sky.
6. Relative positions between objects: In Image 1, objects are placed on a table, while in Image 2, objects are positioned along a road or street.
Question: Are there people in both images?
Answer: Yes, there are people in both images.
Question: what is the difference in the scene of the two images?
Answer: The scene in Image 1 is set in an indoor or semi-enclosed space, such as a café or restaurant, while Image 2 is set outdoors, in a cityscape or urban environment.
Question: What objects are present in Image 1 but not in Image 2?
Answer: Image 1 contains objects like a glass, bottle, and chairs, which are not present in Image 2.
Question: What objects are present in Image 2 but not in Image 1?
Answer: Image 2 has elements like a bridge, wall, and divider cones, which are not in Image 1.
Question: What is the difference between people in the two images?
Answer: In Image 1, In Image 1, there are two women taking a picture, while in Image 2, there is a single person standing near a bus.
Question: What are the people doing in the two images?
Answer: The people in Image 1 are taking a picture, while the person in Image 2 is not engaged in any specific action.
Question: What is the difference in the locations of the two images?
Answer: The scene in Image 1 is more confined, with a table and chairs, while Image 2 has a wider view, including a road, buildings, and the sky.
Question: What is the difference in the relative positions between objects in the two images?
Answer: In Image 1, objects are placed on a table, while in Image 2, objects are positioned along a road or street.

Image 1: <caption1>
Image 2: <caption2>
Summary: <summary>
Question:

Figure 17: Prompt for the task of generating Q&A pairs based on both image captions and the CaD summary.

**System prompt:**
You are an intelligent chatbot designed for evaluating the correctness of generative outputs for question-answer pairs. Your task is to compare the predicted answer with the correct answer and determine if they match meaningfully. Here's how you can accomplish the task:

##INSTRUCTIONS:
- Focus on the meaningful match between the predicted answer and the correct answer.
- Consider synonyms or paraphrases as valid matches.
- Evaluate the correctness of the prediction compared to the answer.

**User prompt:**
Please evaluate the following question-answer pair:
Question: <question>
Correct Answer: <answer>
Predicted Answer: <prediction>
Evaluate if the predicted answer is correct with yes/no and assign a correctness score between 0 and 5, where 0 indicates incorrect answer, and 5 signifies the highest meaningful match. Please generate the response in the form of a Python dictionary string with keys 'pred' and 'score', where value of 'pred' is a string of 'yes' or 'no' and value of 'score' is in INTEGER, not STRING. DO NOT PROVIDE ANY OTHER OUTPUT TEXT OR EXPLANATION. Only provide the Python dictionary string. For example, your response should look like this: {'pred': 'no', 'score': 0}.

Figure 18: Prompt for the LLM-assisted evaluation.

## F  LIST OF ASSETS

Our image sources and annotations are obtained from public datasets. We release our data in accordance to the source data licenses.

Here is a list of image sources:

- Open Images v6 (Kuznetsova et al., 2020) (`https://storage.googleapis.com/openimages/web/download_v6.html`): The images are under Creative Commons Attribution (CC BY) 2.0 license.

- COCO 2017 (Chen et al., 2015; Lin et al., 2014) (`https://cocodataset.org/#download`): The images are under a Creative Commons Attribution 4.0 license.

- Flicker30K (Young et al., 2014) (`https://shannon.cs.illinois.edu/DenotationGraph/`): The images are the property of SmugMug or its third party licensors and are protected by United States and international intellectual property laws. The images are provided for researchers and educators who wish to use the dataset for non-commercial research and/or educational purposes.

- ADE20K (Zhou et al., 2019) (`https://groups.csail.mit.edu/vision/datasets/ADE20K/index.html#Download`): The images belong to MIT CSAIL and are licensed under a Creative Common BSD-3 License.

- Visual Genome (Krishna et al., 2017) (`https://homes.cs.washington.edu/~ranjay/visualgenome/api.html`): The images are under a Creative Commons Attribution 4.0 license.

Here is a list of image annotation sources:

- Localized narratives (Pont-Tuset et al., 2020) (`https://google.github.io/localized-narratives/`): The annotations are released under a Creative Common Attribution (CC BY) 4.0 license.

- MIMIC-IT (Li et al., 2023a) (`https://huggingface.co/datasets/pufanyi/MIMICIT`): The annotations are released under an MIT license.

- SVIT (Zhao et al., 2023) (`https://huggingface.co/datasets/BAAI/SVIT`): The annotations are licensed under a Creative Commons Attribution 4.0 license. It should abide by the policy of OpenAI (`https://openai.com/policies/terms-of-use`). The use of original images and annotations from Visual Genome and MS-COCO should comply with the original licenses.

Here is a list of implementation sources or model weights:

- LLaVA (Liu et al., 2023b;a) (`https://github.com/haotian-liu/LLaVA`): The code is released under an Apache-2.0 license. The project utilizes certain datasets and checkpoints that are subject to their respective original licenses, including but not limited to the OpenAI Terms of Use[3] for the dataset and the specific licenses for base language models for checkpoints trained using the dataset (*e.g.* LLaMA community license[4] for LLaMA-2 and Vicuna-v1.5).

- Mixtral 8×7B model (Jiang et al., 2024) (`https://huggingface.co/mistralai/Mixtral-8x7B-v0.1`): The model is released under an Apache-2.0 license. Usage is subject to the term of use for Mistral products and services[5].

---

[3]`https://openai.com/policies/eu-terms-of-use/`
[4]`https://ai.meta.com/llama/license/`
[5]`https://mistral.ai/terms/#terms-of-use`