# OpenReview forum: "Comparison Visual Instruction Tuning"
_ICLR.cc/2025/Conference — ICLR 2025 Conference Withdrawn Submission_

### Official Review · Reviewer_6sP9 · 2024-10-31

**Soundness:** 3
**Presentation:** 3
**Contribution:** 3
**Rating:** 5
**Confidence:** 4

**Summary:**

The paper proposes CaD-VI, a novel two-phase approach to improve large multimodal models (LMMs) in spotting and reasoning about commonalities and differences (CaD) between images. The authors introduce CaD-Inst, a dataset containing 349K image pairs with synthetic CaD instructions generated through CaD-VI. They demonstrate that training on CaD-Inst significantly enhances CaD reasoning in LMMs, advancing performance by up to 17.5% on related tasks. Additionally, the authors contribute CaD-QA, a benchmark with 7.5K questions for open-ended CaD evaluations, showcasing substantial improvements over existing state-of-the-art models​

**Strengths:**

1. The paper introduces a unique two-phase methodology, CaD-VI, specifically designed for training LMMs on commonality and difference (CaD) tasks. This novel approach effectively fills a gap in multimodal AI, where CaD reasoning has received limited focus.
2. By creating and releasing the CaD-Inst dataset with 349K image pairs, the authors provide a valuable resource for training and evaluating LMMs on nuanced visual reasoning tasks. This large dataset enhances model robustness in spotting both differences and similarities between images, a significant step beyond traditional difference-only datasets.
3. The model trained on CaD-Inst shows substantial performance gains, improving up to 17.5% over current state-of-the-art models on CaD-related tasks. This demonstrates the practical effectiveness of the dataset and methodology in advancing LMM performance.

**Weaknesses:**

1. The paper’s experiments mainly focus on benchmark performance gains without a clear demonstration of how the model performs on real-world, uncurated image pairs. Maybe like LVLM benchmarks like MME, hallusionbench, MMMU, MMC-Benchmark and so on. In hallusionbench, MMMU and MMC-Benchmark, they also have multiple image as input.
2. The authors mentioned that they use Mixtral 8 x 7B as the evaluator. How does it compare with human evaluation and GPT4? Is it possible to use non-llm method?
3. Is It possible for the authors to compare with more recently MLLMs like internvl, Cambrain and Eagle?

**Questions:**

My questions are mentioned in the weakness.

---

### Official Review · Reviewer_7j3W · 2024-10-31

**Soundness:** 3
**Presentation:** 2
**Contribution:** 2
**Rating:** 3
**Confidence:** 4

**Summary:**

This paper focuses on a specialized task within multi-image question answering: comparing two images to identify their Commonalities and Differences (CaD). The authors propose a two-phase pipeline to generate CaD-formatted data and to train specific CaD-focused LMMs. To evaluate the model’s effectiveness in solving CaD problems, a targeted evaluation benchmark was proposed. Extensive experiments demonstrate that the proposed dataset and methodology significantly enhance the model's performance on CaD-related tasks.

**Strengths:**

1. This paper addresses an overlooked sub-problem in multi-image question answering. The proposed new dataset and benchmark may support the development of versatile multimodal large models and facilitate comprehensive evaluations.
2. Detailed ablation experiments on the data construction method provide valuable insights, guiding future training data development for multimodal models.

**Weaknesses:**

1. The paper’s scope is limited, focusing narrowly on comparing two images in terms of commonalities and differences (CaD), a sub-capability within the broader multi-image question-answering domain. Its main contribution lies in using LLMs to generate data for CaD, fine-tuning existing VLMs on this data, and performing in-domain evaluations, which constrains its general applicability.
2. The two-phase pipeline is similar to the multi-stage annotation process introduced by Kirillov et al.[1] However, unlike in [1], where extensive manual efforts and model confidence checks ensure data quality, this paper’s Phase 2 lacks additional quality assurance measures. Consequently, CaDv2 (71K) may lack reliability, functioning merely as unlabeled data for self-distillation.
3. Following from point 2, it is misleading for the authors to label CaDv1 and CaDv2 collectively as the CaD-Inst dataset. The authors should clarify the distinctions between the proposed dataset and training method to avoid confusion.

Reference:
[1] Kirillov A, Mintun E, Ravi N, et al. Segment anything. Proceedings of the IEEE/CVF International Conference on Computer Vision, 2023: 4015-4026.

**Questions:**

N/A

---

### Official Review · Reviewer_XHGB · 2024-11-03

**Soundness:** 3
**Presentation:** 2
**Contribution:** 2
**Rating:** 3
**Confidence:** 5

**Summary:**

This paper introduces a two-phase approach for visual instruction data collection, emphasizing the significance of identifying commonalities and differences in image pairs. The authors argue that understanding these aspects is crucial to improving large multimodal model (LMM) performance.

The proposed methodology is as follows:

- **Phase 1:** Image pairs with captions are initially gathered, followed by prompting a large language model (LLM) to describe their commonalities and differences. This synthetic data is then utilized for instruction tuning, resulting in a baseline instruction-tuned LLaVA variant model.
- **Phase 2:** The model trained in Phase 1 is employed as an annotator to generate additional data, effectively expanding the instruction tuning dataset for further refinement of the LLaVA model.

**Strengths:**

- The concept of recognizing commonalities and differences aligns well with human visual perception and design principles, bringing an innovative perspective to visual data processing.
- The two-phase strategy is technically sound and shows promise for enhancing LMM capabilities.
- Comprehensive statistics and data sources are provided, which strengthens the transparency of the data collection process.
- Implementation details are thoroughly documented, offering useful insights for replication.
- The experimental section includes comparisons with recent models like LLaVA 1.6, 1.5, and InternLM, giving context to the model's relative performance.

**Weaknesses:**

- The paper offers limited algorithmic innovation, as much of the method depends on synthetic data generation without addressing potential pitfalls.
- The quality of generated data is uncertain, particularly regarding hallucinations or inaccuracies in synthetic outputs. More clarity on quality control measures or validation processes is needed to ensure data reliability.
- The effectiveness of the commonalities and differences data remains ambiguous. In Table 7, results are presented with varied training data sizes, complicating a fair comparison. The authors should consider experiments with controlled dataset sizes across conditions, allowing for a clearer assessment of data type impact rather than quantity alone.
- The manuscript contains several writing issues that hinder comprehension, particularly in the abstract and conclusion sections. A clearer, more concise explanation of the methodology and results would improve readability.

**Questions:**

- How does the model behave when scaling the quantity of commonalities and differences data separately? An investigation into scaling laws for this approach could be insightful.
- What constitutes an ideal or poor image pair for this method? Does random image pairing suffice, or are there specific factors to consider for high-quality data collection?

---

### Official Review · Reviewer_H34Y · 2024-11-03

**Soundness:** 3
**Presentation:** 3
**Contribution:** 2
**Rating:** 5
**Confidence:** 3

**Summary:**

the focus of the paper is to improve the model's understanding capability in terms of Commonalities and Differences (CaD) over two images. to address the problem, the paper proposes a two-stage approach, in which the first stage utilizes LLM to generate the CaD information based on the caption only from the Localized Narratives dataset and then train a model. the second stage applies the trained model to generate more training data. a QA dataset is created to further diversity and enhance the data source.

**Strengths:**

this paper introduces CaD-VI, a two-phase approach for enhancing visual instruction tuning in large multimodal models (LMMs) with a specific focus on comparing commonalities and differences (CaD) between image pairs. this work addresses an underexplored area in LMMs and providing valuable insights for visual reasoning.

the paper contributes a dataset CaD-Inst containing 349K image pairs for CaD instruction tuning, and CaD-QA, a benchmark of 7.5K open-ended questions designed to evaluate CaD capabilities. These resources are likely to foster advancements in LMM training and evaluation.

**Weaknesses:**

the paper uses the open-sourced LLM or MLLM to generate the training data. what if using the GPT-4 model to generate data directly from the image, rather than from the image descriptions? one less-costly approach could be just to test, say, 5 data points, and see how the gap is, although a more comprehensive way is to replace the open-sourced component with, e.g. gpt-4 in the proposed pipeline. in this way, it may also work to directly use the image rather than the caption to generate the CaD data in the first stage?

in the proposed approach, there are two stages. the first is to generate training data and then do the training on these data. the second is to use the trained model to generate the training data. what if skipping the second phase, but generating more data in the first stage? this sounds like more reasonable and easier, maybe also more effective, way? one reason of why the second stage exists could be to save the cost, as the first stage requires the captioning and LLM to generate data, but this may not be a problem since all components are open-sourced?

in the first stage, the caption is from the dataset itself (this may also be part of the reason why stage 2 exists).  what if applying a MLLM to do the captioning? if it also works, it would make the pipeline more scalable to generate even more data.

another question is how the scaling curve looks like with more training data in the first stage. that is, if it is 1k, how the perf is; if it is 10k, how the perf is; if it is 100k, .... the reason i'm more interested in this question is that the paper basically curates some training data and then fine-tune the model on these data to improve the performance on the target domain. this is a quite reasonable approach. the dataset scale here could be one of the main factors that impact the performance.

**Questions:**

see weakness.

---

### Note · Authors · 2024-11-25

**Comment:**

After careful consideration, we have decided to withdraw our submission. However, we would like to address some of the key concerns raised by the reviewers. We that our clarifications provide greater insight into the rationale and robustness of our approach.

>Two-stage pipeline

In this work, we focus on employing open-source large foundation models for data collection. The current open-source Multimodal LLMs do not have strong enough capabilities of visual reasoning and instruction following when processing multiple input images, as most of their training data contains only one input image. In this case, using caption as a “symbolic” representation of each image and employing an LLM with strong text instruction-following ability for generation of comparison summary of multiple input images is a more robust way of data collection than using open-source MLLMs. The practice of this data collection pipeline is verified in the original work of LLaVA. Additionally, in the second stage of our approach - we generate CaD data leveraging both captions and the CaD image analysis capabilities of our stage-1 model. This significantly reduces hallucinations and improves the quality of the 2nd stage CaD dataset as evident by the significant performance improvement obtained by our stage-2 model over stage-1 model (Table 5, rows E vs F, in the paper).

>Usage of GPT-4 model

We argue that the usage of GPT4o for data collection is a good option but orthogonal to our approach. It can be added to any stage in our approach.
However, there are two factors to keep in mind
1) using GPT4o might prevent free commercial use. Due to the GPT license constraint, training models with GPT data for enterprise use cases requires special agreement with OpenAI.
2) GPT4o does not always have strong instruction-following ability in the case of input of multiple images. In Figure 2(a) in the attached general response PDF, the answer from GPT4o contains descriptions of both images while only image 2 is asked in the question. The description of “black pants” is also missed in the answer.

>Other LLM-Assisted Evaluation and Human Evaluation

To evaluate the impact of different LLMs on LLM-assisted evaluations, we employed LLaMA 3.1 70B and GPT4o Mini for the assessment of CaD-QA. Additionally, we conducted human evaluations by randomly sampling 150 open-ended questions from the benchmark and asked three volunteers to rate (on a scale of 0–5) the predictions of MLLMs. The results demonstrated a clear human preference for answers from CaD-VI, supporting our choice of LLMs.

>Usage of Synthetic captions

We agree that synthetic captions generated by open-source models can improve summary quality. In our ablation study, we demonstrated that adding image captions in the second stage yields better results. By leveraging the stage-1 model, we were able to cross-reference synthetic captions with the original images, enriching the data further.
It is worth noting that while synthetic captions are individually generated, they may lack some fine-grained CaD-related details. However, our stage-1 model successfully addresses these gaps during the second stage of the pipeline, as shown in Table 6.

>General Vision-Language (VL) Benchmarks

In addition to the CaD datasets, we reported performance on general VL benchmarks, including SEED-Bench (image and video). Furthermore, we compared the performance of CaD-VI against the original LLaVA model on MME and MMBench, as detailed in the appendix.

**Withdrawal Confirmation:**

I have read and agree with the venue's withdrawal policy on behalf of myself and my co-authors.